# MARK2 regulates Golgi apparatus reorientation by phosphorylation of CAMSAP2 in directional cell migratio

**Peipei Xu[1,2†], Rui Zhang[1†], Zhengrong Zhou[1,3†], Honglin Xu[1], Yuejia Li[1,2], Mengge Yang[1,2], Ruifan Lin[4], Yingchun Wang[1,2,5], Xiahe Huang[1], Qi Xie[4], Wenxiang Meng[1,2,5]\***

[1]State Key Laboratory of Molecular Developmental Biology, Institute of Genetics and Developmental Biology, Chinese Academy of Sciences, Beijing, China; [2]University of Chinese Academy of Sciences, Beijing, China; [3]Neuroscience Center, Department of Basic Medical Sciences, Shantou University Medical College, Shantou, China; [4]Wangjing Hospital of China Academy of Chinese Medical Sciences, Beijing, China; [5]Innovation Academy for Seed Design, Chinese Academy of Sciences, Beijing, China

## eLife Assessment

The authors propose that the kinase MARK2 regulates the Golgi's reorientation towards the cell's leading edge through the regulation of microtubule binding protein CAMSAP2 and its binding to USO1. While the model is interesting and the study is **useful**, the quantification of an insufficient number of cells and insufficient description of the methods and biological replicates mean the results are **inadequate** to support the model.

[Editors' note: this paper was reviewed by Review Commons.]

**\*For correspondence:**
wxmeng@genetics.ac.cn

[†]These authors contributed equally to this work

**Competing interest:** The authors declare that no competing interests exist.

**Abstract** The reorientation of the Golgi apparatus is crucial for cell migration and is regulated by multipolarity signals. A number of non-centrosomal microtubules anchor at the surface of the Golgi apparatus and play a vital role in the Golgi reorientation, but how the Golgi are regulated by polarity signals remains unclear. Calmodulin-regulated spectrin-associated protein 2 (CAMSAP2) is a protein that anchors microtubules to the Golgi, a cellular organelle. Our research indicates that CAMSAP2 is dynamically localized at the Golgi during its reorientation processing. Further research shows that CAMSAP2 is potentially regulated by a polarity signaling molecule called MARK2, which interacts with CAMSAP2. We used mass spectrometry to find that MARK2 phosphorylates CAMSAP2 at serine-835, which affects its interaction with the Golgi-associated protein USO1 but not with CG-NAP or CLASPs. This interaction is critical for anchoring microtubules to the Golgi during cell migration, altering microtubule polarity distribution, and aiding Golgi reorientation. Our study reveals an important signaling pathway in Golgi reorientation during cell migration, which can provide insights for research in cancer cell migration, immune response, and targeted drug development.

## Introduction

Cell migration plays a pivotal role in numerous physiological processes, including embryonic development, tissue regeneration, and immune response mechanisms (*Ridley et al., 2003*; *Jaffe and Hall, 2005*). This intricate process is significantly influenced by the dynamic reorientation of the Golgi apparatus, which orchestrates the targeted delivery of newly synthesized membranes and proteins

to the leading edge of the cell (*Mellor, 2004*; *Yadav and Linstedt, 2011*). As early as 1970, scientists discovered that the chick corneal epithelium underwent apical and basal transitions of the Golgi during development and proposed that there must be signaling pathways that directly regulate Golgi reorientation. However, the molecular mechanisms that regulate Golgi reorientation in cells are still unclear.

In interphase cells, the Golgi apparatus typically occupies a central position. However, during migration, it undergoes a strategic repositioning toward the migration front (*Millarte and Farhan, 2012*; *Meiring et al., 2020*). Golgi apparatus becomes a dispersed, unpolarized transitional structure, then transforming into a complete Golgi apparatus with polarity, intricately tied to the microtubule networks (*Yadav and Linstedt, 2011*; *Martin et al., 2018*). Interruptions in this network, e.g., by nocodazole, disrupt the Golgi structure, impeding its necessary reorientation (*Thyberg and Moskalewski, 1999*). Non-centrosomal microtubules, rooted in the Golgi, stabilize their ribbon structure and facilitate this reorientation (*Akhmanova and Steinmetz, 2019*). CAMSAP2, a microtubule minus-end binding protein, emerges as a crucial player in this context (*Tanaka et al., 2012*; *Wu et al., 2016*; *Akhmanova and Steinmetz, 2019*). Diminished CAMSAP2 levels lead to a loss of Golgi-associated non-centrosomal microtubules, disrupting Golgi function and hampering cell migration (*Tanaka et al., 2012*; *Wu et al., 2016*; *Akhmanova and Steinmetz, 2019*; *Ravichandran et al., 2020*). This emphasizes CAMSAP2's role in sustaining Golgi integrity during critical cellular events like migration.

Cell migration is a complex process that relies on intricate signaling pathways and the coordination of various cellular components, such as the microtubule networks and Golgi apparatus (*Preisinger et al., 2004*; *Bisel et al., 2008*; *Etienne-Manneville, 2008*; *Matsui et al., 2015*; *Mardakheh et al., 2016*). Rho GTPases, including Rho, Rac, and Cdc42, are instrumental in the cytoskeletal reorganization needed for migration (*Iden and Collard, 2008*). They govern the formation of stress fibers and the protrusion of lamellipodia and filopodia, all underpinned by the microtubule network (*Garcin and Straube, 2019*; *SenGupta et al., 2021*). Microtubule stability, essential for cell polarity and movement, can be modulated by protein kinases such as GSK-3β and the PAR6-PAR3-aPKC complex (*Suzuki and Ohno, 2006*; *Goldstein and Macara, 2007*; *Etienne-Manneville, 2008*; *Iden and Collard, 2008*). Motor proteins associated with microtubules, namely dynein and kinesin, are involved in the positioning and transport of the Golgi (*Etienne-Manneville and Hall, 2001*; *Etienne-Manneville and Hall, 2003*; *Miserey-Lenkei et al., 2017*). Additionally, Rab GTPases regulate Golgi trafficking to ensure the delivery of essential components for cell polarization and movement (*Barr, 2009*; *Liu and Storrie, 2012*). ERK and integrin signaling are also implicated in aligning the Golgi along the cell's axis of migration to facilitate polarized secretion (*Millarte and Farhan, 2012*). Despite the understanding of these pathways in regulating microtubule and Golgi dynamics, the precise mechanisms of their coordination during cell migration remain an area of active investigation.

Directional cell migration refers to the directional movement of cells through cytosolic deformation, which is constituted by the following five steps in close spatial and temporal coordination: the cell is stimulated by intracellular and extracellular signals, the cytosol extends the lamellar pseudopods forward, the Golgi transforms into a dispersed, unpolarized excess, followed by the formation of a complete polar Golgi body to complete the Golgi reorientation as well as the retraction of the tails. In this study, we focus on the Golgi reorientation: the process by which dispersed unpolarized Golgi stacks form a completely polarized Golgi. We found that the co-localization of CAMSAP2 with the Golgi dynamically changed during directional cell migration, suggesting that CAMSAP2 is dynamically regulated by polarity signaling. Additionally, immunoprecipitation confirmed the interaction between MARK2 and CAMSAP2. Mass spectrometry found that MARK2 phosphorylates CAMSAP2 at serine-835, which affected its interaction with the Golgi-associated protein USO1 but not with CG-NAP or CLASPs. Here, we reveal a novel polarity signaling pathway: MARK2 acts as a polarity-establishing key kinase that regulates Golgi reorientation by phosphorylating serine at position 835 of CASMAP2, thereby affecting CAMSAP2's binding ability to Golgi-associated protein USO1.

## Results

### Dynamic regulation of the CAMSAP2-Golgi relationship during Golgi reorientation

To investigate the role of non-centrosomal microtubules in the Golgi apparatus reorientation process, we utilized the microtubule minus-end protein CAMSAP2 as a starting point. CAMSAP2 functions by anchoring the minus ends of non-centrosomal microtubules to the Golgi apparatus, facilitating the formation of ribbon structures (*Tanaka et al., 2012*; *Wu et al., 2016*; *Akhmanova and Steinmetz, 2019*; *Ravichandran et al., 2020*). We established a CAMSAP2 knockout cell line in HT1080 cells, a human fibrosarcoma cell line widely employed in cell migration studies (*Rasheed et al., 1974*; *Figure 1—figure supplement 1A and B*). Compared to normal cells, CAMSAP2 knockout cells exhibited disordered Golgi apparatus structure, characterized by dispersion into numerous small stack structures (*Figure 1—figure supplement 1C and D*). However, immunoblotting analysis revealed no change in the protein level of GM130 (*Figure 1—figure supplement 1E*). These findings suggest that CAMSAP2 plays a crucial role in regulating Golgi structure and morphology, consistent with prior research (*Wu et al., 2016*; *Martin et al., 2018*), thus affirming the success of generating CAMSAP2 knockout HT1080 cells.

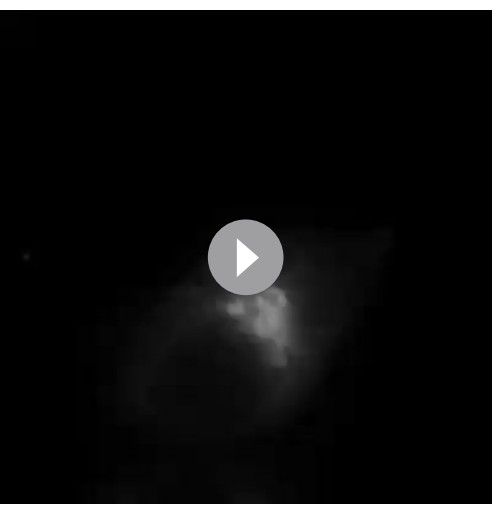

**Video 1.** Live-cell imaging to monitor the Golgi reorientation process.
https://elifesciences.org/articles/105977/figures#video1

Subsequently, to confirm CAMSAP2's role in cell migration and Golgi apparatus reorientation, we conducted a wound healing assay to induce directional cell migration. Previous studies on cell wound healing have demonstrated that as early as 2 hr post-scratch, Golgi reorientation is observed in cells at the leading edge of the wound (where a polarized Golgi apparatus, situated at the front edge of the nucleus, aligns toward the scratch side). We fixed the cells after 2 hr of migration with methanol for staining and assessed Golgi reorientation by measuring the angle between the Golgi centroid-nucleus centroid axis and the direction of cell migration. If the absolute value of this angle is within 90 degrees, it indicates that the Golgi is positioned at the front edge of the nucleus and oriented in the migration direction, signifying proper Golgi reorientation. Conversely, if the absolute angle falls between 90 and 180 degrees, it suggests that the Golgi is positioned behind the nucleus, indicating incomplete Golgi reorientation. Compared to control HT1080 cells, the proportion of cells with completed Golgi reorientation was significantly reduced in CAMSAP2 knockout HT1080 cells (*Figure 1—figure supplement 1F–H*). To verify the role of CAMSAP2 in directed migration of HT1080 cells, we conducted a scratch assay. Bright-field images were taken of the cells before and 8 hr after scratch formation to measure the change in scratch area over this period. The results showed a significant reduction in migration speed in CAMSAP2 knockout HT1080 cells compared to control cells (*Figure 1—figure supplement 1I and J*). These findings strongly suggest CAMSAP2's involvement in the Golgi apparatus reorientation process, aligning with previous studies (*Martin et al., 2018*).

Next, we observed whether Golgi non-centrosomal microtubules undergo dynamic regulation during Golgi reorientation. First, we used live-cell imaging to monitor the Golgi reorientation process and observed that the Golgi apparatus transitions into a dispersed intermediate structure before restoring its complete structure (*Figure 1—figure supplement 2A* and *Video 1*). CAMSAP2 is responsible for anchoring the minus ends of non-centrosomal microtubules to the Golgi apparatus. We performed fixation and staining at different time points in the cell wound healing assay. By analyzing CAMSAP2 localization in different states, we found that CAMSAP2 co-localization on the dispersed intermediate Golgi structure was significantly lower than on the fully intact Golgi (*Figure 1A and B*). The pattern of CAMSAP2 localization on the Golgi apparatus in the leading-edge cells at different time points in the wound healing assay is shown in *Figure 1—figure supplement 2B and C*. Since in wound healing assays, the signaling for directed migration is gradually transmitted from the cells at

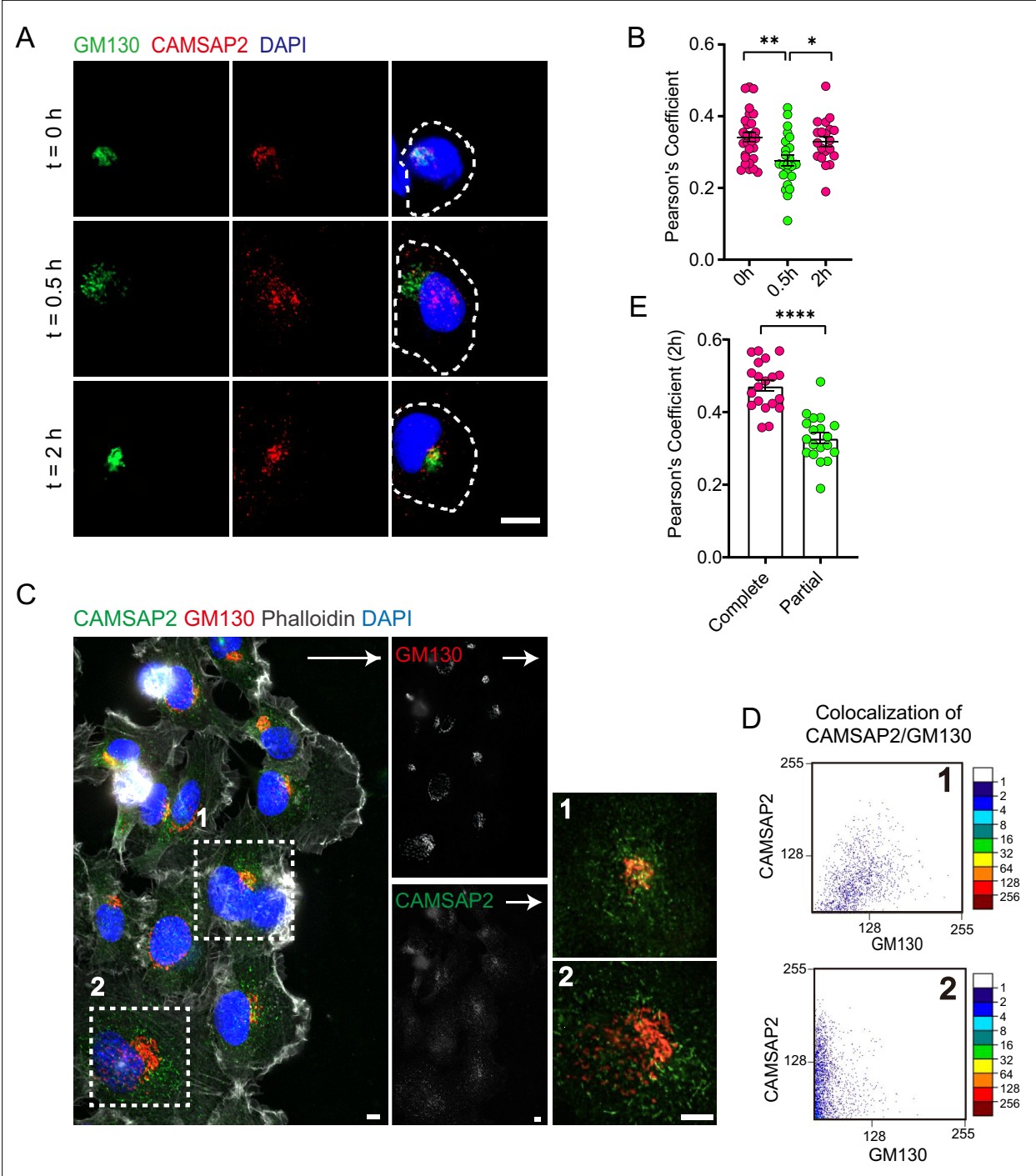

**Figure 1.** CAMSAP2 is dynamically localized at the Golgi during the reorientation process in directional cell migration. (**A**) Immunofluorescence staining of HT1080 located at the foremost edge of the cell wound after 0, 0.5, and 2 hr migration. White arrows represent the direction of cell migration. Shown are representative immunofluorescence staining images. Scale bar = 10 μm. (**B**) Quantification of the localization of CAMSAP2 on the Golgi after 0, 0.5, and 2 hr migration. Each symbol represents an individual cell. The co-localization of representative immunofluorescence staining plots corresponding to each time point is counted. Values = means ± SEM, unpaired two-tailed Student's *t*-test, \*\*p<0.01 (N=3; 0 hr: n=28; 0.5 hr: n=23; 2 hr: n=20). (**C**) Immunofluorescence staining of HT1080 after 2 hr of migration in wound healing assay. The cell in the white dashed box shows a different cell state. The cell in the white dashed box on the 1st is the cell closest to the edge of the wound and has already completed the process of directional cell migration: the Golgi is polar and is located at the anterior edge of the nucleus. At this point, CAMSAP2 is more localized to the Golgi. The cells in the white dashed box on the 2nd have not yet completed the process of cell-directed migration: the Golgi is relatively dispersed and has lost its polarity. At this point, the localization of CAMSAP2 on the Golgi is substantially reduced. Golgi was stained with mouse anti-GM130 antibody (red). Nuclei were stained

*Figure 1 continued on next page*

*Figure 1 continued*

with DAPI (blue). CAMSAP2 was stained with rabbit anti-CAMSAP2 antibody (green). On the right is a magnified image of the white dashed box. Scale bars = 5 µm. White arrows represent migration direction. (**D**) Co-localization of CAMSAP2 and GM130 was analyzed using the ScatterJ plug-in in ImageJ software. The horizontal and vertical axes of the scatterplot represent the gray values obtained for each pixel point in each channel, and the closer the image is to the diagonal, the higher the degree of co-localization is indicated. (**E**) Quantification of the localization of CAMSAP2 on the Golgi in different cell states in (**C**). Each symbol represents an individual cell. Values = means ± SEM, unpaired two-tailed Student's *t*-test, ***p<0.001 (N=3; complete: n=20; partial: n=19).

The online version of this article includes the following source data and figure supplement(s) for figure 1:

**Source data 1.** Excel file containing the quantified data of statistic analysis for *Figure 1B and E*.

**Figure supplement 1.** CAMSAP2 is essential for Golgi reorientation during directed migration.

**Figure supplement 1—source data 1.** PDF containing original scans of the relevant western blot analysis with highlighted bands and sample labels for *Figure 1—figure supplement 1B*.

**Figure supplement 1—source data 2.** Original file for the western blot analysis in *Figure 1—figure supplement 1B*.

**Figure supplement 1—source data 3.** Excel file containing the quantified data of statistic analysis for *Figure 1—figure supplement 1D, E, G, H, and J*.

**Figure supplement 2.** CAMSAP2 is dynamically localized at the Golgi during the reorientation process in directional cell migration.

the leading edge of the scratch to the inner cells, we examined immunostaining images taken 2 hr after scratching. Specifically, we observed the localization of CAMSAP2 on the Golgi apparatus in the leading-edge cells compared to the inner cells. We found that the localization of CAMSAP2 on the dispersed Golgi apparatus in the inner cells was significantly lower than its localization on the intact Golgi apparatus in the leading-edge cells (*Figure 1C–E*). We carefully examined the nuclear morphology of dispersed Golgi cells and found no evidence of mitotic features, indicating that these cells are not undergoing division. Furthermore, during the scratch wound assay, we use 2% serum to culture the cells, which helps minimize the impact of cell division. Our experiments unveil a previously unrecognized phenomenon: dynamic changes in CAMSAP2 localization at the Golgi during cell-directed migration, suggesting the dynamic regulation of non-centrosomal microtubules during Golgi reorientation.

## MARK2 regulates CAMSAP2 distribution at Golgi apparatus

The role of non-centrosomal microtubule networks in the Golgi apparatus reorientation and their involvement in intracellular and extracellular polarity signal transduction pathways remains largely unknown. To identify potential signaling pathways that regulate the reorientation of the Golgi apparatus through non-centrosomal microtubules, we screened for possible candidate proteins based on existing data: these polarity regulation signals are involved in the Golgi apparatus and associated with CAMSAP2 (*Wu et al., 2016*; *Fasimoye et al., 2023*). We first conducted a KEGG pathway analysis of all proteins potentially localized at the Golgi apparatus, selecting those capable of establishing and maintaining cell polarity (*Figure 2A*). These were then cross-referenced with proteins reported in the literature to interact with CAMSAP2 (*Figure 2B*). Among the selected proteins, we focused on the key polarity protein MARK2, which regulates cell migration and the differentiation of axons and dendrites (*Sapir et al., 2008*; *Nishimura et al., 2012*; *Zhou et al., 2020*; *Pasapera et al., 2022*). We confirmed the interaction between CAMSAP2 and MARK2 through co-immunoprecipitation experiments (*Figure 2C*). Through immunoprecipitation experiments, we found that CAMSAP2 interacts with MARK2 via its 1449F terminal domain (*Figure 2—figure supplement 1A and B*). To investigate whether MARK2 influences cell migration and Golgi reorientation, we established a MARK2 gene knockout HT1080 cell line using CRISPR/Cas9; the absence of MARK2 can also influence the orientation of the Golgi apparatus during cell wound healing and cause a delay in wound closure (*Figures 2D–I and 3D*). In MARK2 knockout cells, overexpression of GFP-MARK2 partially restores Golgi structure and rescues the reduced CAMSAP2 localization to the Golgi (*Figure 2—figure supplement 1C–E*).

Next, we investigated whether MARK2 influences Golgi apparatus morphology by regulating CAMSAP2. Immunofluorescence staining analysis revealed that the Golgi apparatus ribbon in the MARK2 knockout cells was disordered and had compromised integrity, with numerous small stack structures (*Figure 3A–B and D–F*). To rule out cell specificity, we observed the same phenomenon

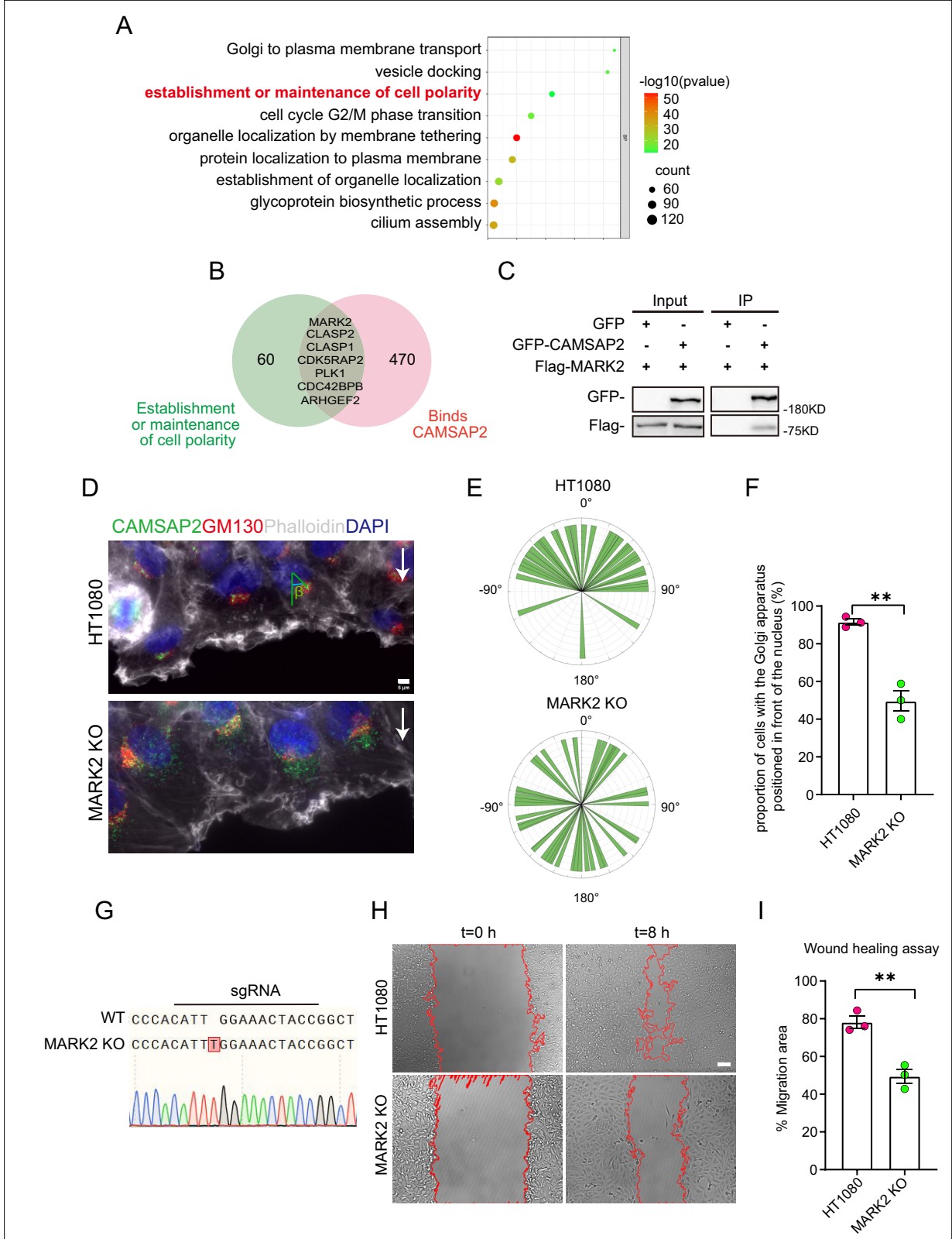

**Figure 2.** Search for polarity-establishing key proteins that can localize to the Golgi and interact with CAMSAP2. (**A**) GO analysis of proteins reported in the literature that may localize on the Golgi (Golgi TAG-IP/Whole-Cell lysate meaningful proteins), and the figure shows the BP (biological process), the protein we focus on that can establish and maintain cell polarity. GO analysis of proteins was plotted by https://www.bioinformatics.com.cn (last accessed on June 20, 2024), an online platform for data analysis and visualization. (**B**) Venn diagram. Proteins in (**A**) that can establish and maintain

*Figure 2 continued on next page*

*Figure 2 continued*

polarity and proteins that may interact with CAMSAP2 were selected for enrichment, and the enriched proteins are displayed in the diagram. The CAMSAP2 interactants were derived from the study by **Wu et al., 2016**, which provides the source of these interactants. (**C**) Immunoblot showing the results of GFP-IP. Immunoprecipitation showing GFP-CAMSAP2 and Flag-MARK2 interactions. The cells used in the experiment were 293T cells. The interaction between CAMSAP2 and MARK2 is referenced from **Zhou et al., 2020**. (**D**) Immunofluorescence staining of HT1080 and MARK2 KO cells in wound healing assay after 2 hr of migration. Golgi was stained with mouse anti-GM130 antibody (red). Nuclei were stained with DAPI (blue). CAMSAP2 was stained with rabbit anti-CAMSAP2 antibody (green). F-actin was stained with phalloidin (white). We fixed the cells after 2 hr of migration with methanol for staining and assessed Golgi reorientation by measuring the angle between the Golgi centroid-nucleus centroid axis and the direction of cell migration. Black arrows point to the direction of migration. Scale bar = 5 μm. (**E**) Images demonstrate cells completing Golgi reorientation. If the absolute value of this angle is within 90 degrees, it indicates that the Golgi is positioned at the front edge of the nucleus and oriented in the migration direction, signifying proper Golgi reorientation. Conversely, if the absolute angle falls between 90 and 180 degrees, it suggests that the Golgi is positioned behind the nucleus, indicating incomplete Golgi reorientation (N=3; HT1080: n=35; MARK2 KO: n=45). (**F**) Quantification of the proportion of cells in (**E**) that can complete Golgi reorientation properly. Values = means ± SEM, unpaired two-tailed Student's *t*-test, **p<0.01 (N=3; HT1080: n=35; MARK2 KO: n=45). (**G**) Sequencing result of the *MARK2* KO cells near the sgRNA sequence. (**H**) Overlay of end-point phase-contrast images with the result of cell tracking after 8 hr of migration in a wound healing assay in control HT1080 and MARK2 KO HT1080 cells. Scale bar = 100 μm. (**I**) Plots show quantification of wound closure area after 8 hr of migration in HT1080 cells and MARK2 KO HT1080 cells. Values = means ± SEM, unpaired two-tailed Student's *t*-test, **p<0.01 (N=3).

The online version of this article includes the following source data and figure supplement(s) for figure 2:

**Source data 1.** PDF containing original scans of the relevant western blot analysis with highlighted bands and sample labels for *Figure 2C*.

**Source data 2.** Original file for the western blot analysis in *Figure 2C*.

**Source data 3.** Excel file containing the quantified data of statistic analysis for *Figure 2E, F, and I*.

**Figure supplement 1.** MARK2 affects CAMSAP2 localization on the Golgi apparatus.

**Figure supplement 1—source data 1.** PDF containing original scans of the relevant western blot analysis with highlighted bands and sample labels for *Figure 2—figure supplement 1B and F*.

**Figure supplement 1—source data 2.** Original file for the western blot analysis in *Figure 2—figure supplement 1B and F*.

**Figure supplement 1—source data 3.** Excel file containing the quantified data of statistic analysis for *Figure 2—figure supplement 1D, E, G, I*.

in RPE1 (*Figure 2—figure supplement 1F–H*). This is consistent with the phenotype of CAMSAP2 knockout cells.

Further observations showed that the knockout of MARK2 affected the localization of CAMSAP2 at the Golgi apparatus. Immunofluorescence co-localization analysis indicated a significant decrease in CAMSAP2 and GM130 co-localization following MARK2 knockout (*Figure 3A and C*). Western blot analysis showed that the knockout of MARK2 did not affect the protein levels of GM130 and CAMSAP2 (*Figure 3D–F*). To rule out cell specificity, we observed the same phenomenon in RPE1 (*Figure 2—figure supplement 1H and I*). The above results show that MARK2 regulates CAMSAP2 distribution at the Golgi apparatus. Subsequently, to confirm these results, we enriched Golgi apparatus components by density gradient differential centrifugation for cell fractionation (*Figure 3G*). Western blotting further confirmed that the proportion of CAMSAP2 was significantly reduced in the Golgi apparatus components in MARK2 knockdown cells (*Figure 3H and I*). In MARK2 knockout cells, overexpression of GFP-MARK2 restores the localization of CAMSAP2 on the Golgi apparatus (*Figure 2—figure supplement 1C–E*). All these data suggest MARK2 regulates the CAMSAP2 distribution at Golgi apparatus.

## MARK2 phosphorylates CAMSAP2 and regulates its localization

MARK2, a serine/threonine protein kinase, plays a crucial role in establishing cell polarity in epithelial and neuronal cells (*Biernat et al., 2002*; *Suzuki et al., 2004*; *Zhou et al., 2020*). It influences microtubule stability by phosphorylating and inactivating several microtubule-associated proteins (*Drewes et al., 1997*; *Schwalbe et al., 2013*). Our research hypothesizes that MARK2's phosphorylation of CAMSAP2 regulates the reorientation of the Golgi apparatus. We initiated this study by purifying GFP-tagged CAMSAP2, GST-tagged MARK2, and GST proteins (*Figure 3—figure supplement 1A and B*). The next step involves using in vitro kinase assays and phosphoproteomic analysis to identify the amino acid residues of CAMSAP2 phosphorylated by MARK2. Based on these findings, we generated mutations at potential phosphorylation sites from serine to alanine, as identified in the mass spectrometry data (*Figure 4A*). Then, co-expressed these mutants with MARK2 in 293T cells. We utilized immunoprecipitation and phospho-tag electrophoresis to evaluate phosphorylation,

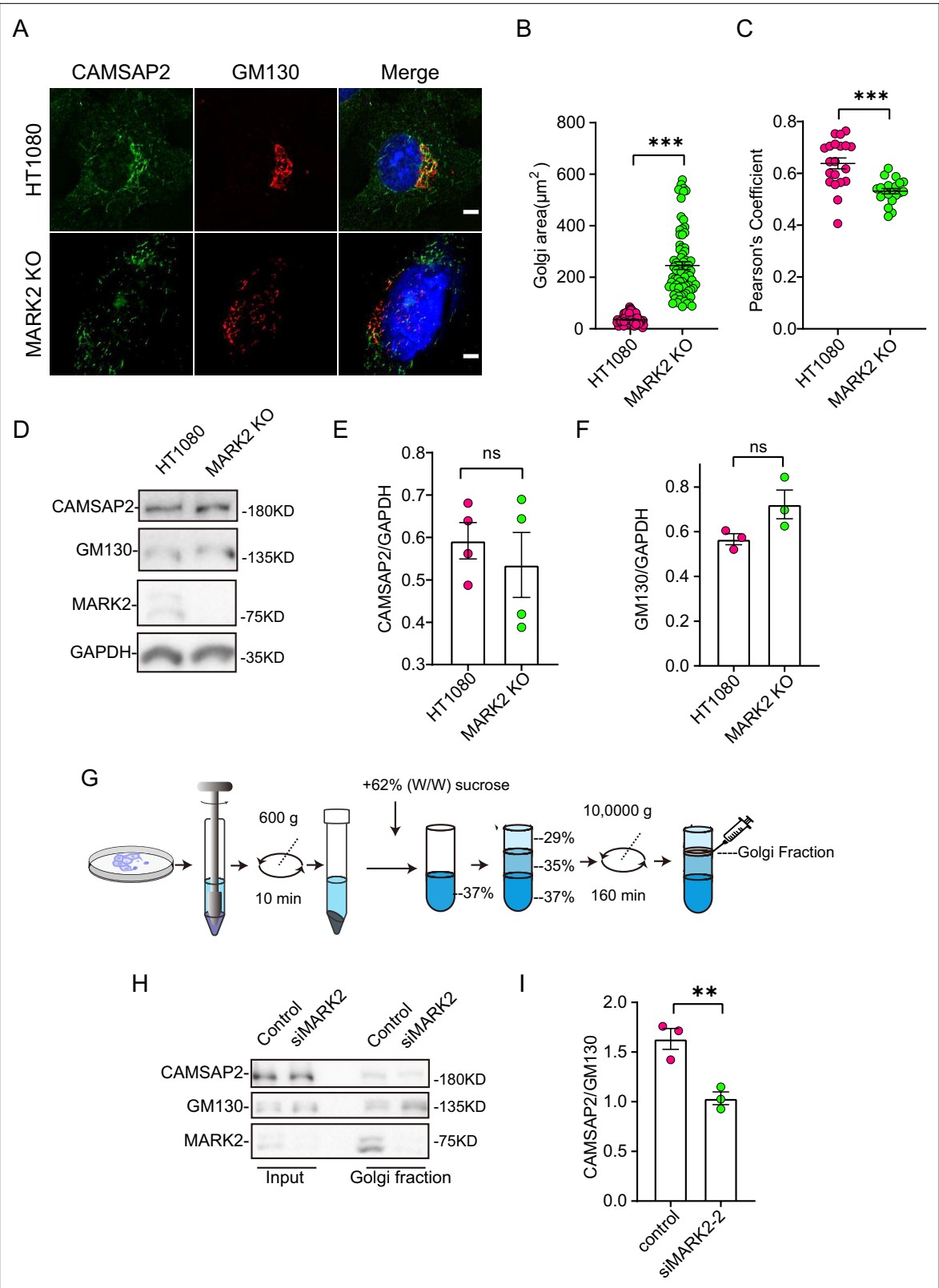

**Figure 3.** MARK2 affects CAMSAP2 localization on the Golgi apparatus. (**A**) Immunostaining of control HT1080 and MARK2 KO cells with rabbit anti-CAMSAP2 (green) and mouse anti-GM130 (red). Nuclei were stained with DAPI (blue). Scale bar = 5 μm. (**B**) Area of the Golgi defined by the Golgi marker GM130 from (**A**). Each symbol represents an individual cell. Values = means ± SEM, unpaired two-tailed Student's *t*-test, ***p<0.001 (N=3; HT1080: n=104; MARK2 KO: n=76). (**C**) ImageJ software was used to calculate CAMSAP2 co-localization with the Golgi marker GM130. Values = means

*Figure 3 continued on next page*

*Figure 3 continued*

± SEM, unpaired two-tailed Student's *t*-test, \*\*\*p<0.001 (N=3; HT1080: n=20; MARK2 KO: n=20). (**D**) Immunoblot of extracts from control HT1080 cells and MARK2 KO cells with rabbit anti-CAMSAP2, mouse anti-GAPDH, or mouse anti-GM130 antibodies. GAPDH was used as a loading control. (**E**) The CAMSAP2/GAPDH ratios from (**D**) were determined using ImageJ software. Values = means ± SEM, unpaired two-tailed Student's *t*-test, p>0.05, N=4. (**F**) The GM130/GAPDH ratios from (**D**) were determined using ImageJ software. Values = means ± SEM, unpaired two-tailed Student's *t*-test, p>0.05, N=3. (**G**) Enrichment of Golgi fraction from control RPE1 cells and MARK2 knockdown cells by density gradient differential centrifugation. (**H**) Immunoblot of homogenate and Golgi fraction from RPE1 cells and MARK2 knockdown cells with rabbit anti-CAMSAP2, rabbit anti-MARK2, or mouse anti-GM130 antibodies. CAMSAP2, GM130, and MARK2 are enriched in the Golgi fraction of RPE1 cells and MARK2 knockdown cells (GM130, Golgi). (**I**) The CAMSAP2/GM130 ratios from (**H**) were determined using ImageJ software. Values = means ± SEM, unpaired two-tailed Student's *t*-test, \*\*p<0.01, N=3.

The online version of this article includes the following source data and figure supplement(s) for figure 3:

**Source data 1.** PDF containing original scans of the relevant western blot analysis with highlighted bands and sample labels for *Figure 3D and H*.

**Source data 2.** Original file for the western blot analysis in *Figure 3D and H*.

**Source data 3.** Excel file containing the quantified data of statistic analysis for *Figure 3B, C, E, F, and I*.

**Figure supplement 1.** MARK2 phosphorylation of serine at position 835 of CAMSAP2.

**Figure supplement 1—source data 1.** PDF containing original scans of the relevant western blot analysis with highlighted bands and sample labels for *Figure 3—figure supplement 1A, B, and C*.

**Figure supplement 1—source data 2.** Original file for the western blot analysis in *Figure 3—figure supplement 1A, B, and C*.

and the result showed that mutating the serine at position 835 in CAMSAP2 inhibited band shifting during electrophoresis, indicating MARK2 phosphorylates CAMSAP2 at this specific serine-835 site (*Figure 4B*, *Figure 3—figure supplement 1C and D*). Furthermore, our comparative sequence analysis of CAMSAP2 across various species showed that the S835 site is highly conserved in mammals and distinctively characteristic of CAMSAP2 within the CAMSAP family (*Figure 4C*), highlighting its potential significance in cellular processes mediated by MARK2.

To investigate the functional implications of MARK2-mediated phosphorylation on CAMSAP2, we transfected HA-CAMSAP2 constructs - wild type (WT), S835A (non-phosphorylatable mutant), and S835D (phosphomimetic mutant) - into CAMSAP2 knockout HT1080 cells (*Figure 4D*). The choice of CAMSAP2 knockout HT1080 cells was made to eliminate potential interference from endogenous CAMSAP2. Given the known disruption of Golgi ribbon structure in the absence of CAMSAP2, our initial focus was on assessing whether these mutants could ameliorate this disruption. Results indicated that the wild-type CAMSAP2 (WT) effectively restored the structural integrity of the Golgi apparatus. In contrast, the non-phosphorylatable (S835A) mutant exhibited a notably reduced ability to restore Golgi structure compared to the phosphomimetic (S835D) mutant (*Figure 4E*). This observation underscores the pivotal role of MARK2-mediated phosphorylation at the S835 site of CAMSAP2 in maintaining Golgi architecture.

Further analysis involved assessing their subcellular localization through immunofluorescence (*Figure 4D*). The WT and the phosphomimetic mutant (S835D) predominantly localized to the Golgi apparatus, whereas the non-phosphorylatable mutant (S835A) manifested as punctate distributions throughout the cytoplasm (*Figure 4F*). These observations indicate that MARK2 phosphorylation of serine at position 835 of CAMSAP2 affects the localization of CAMSAP2 on the Golgi and regulates Golgi structure.

## The phosphorylation of CAMSAP2 at S835 does not affect its interactions with CG-NAP or CLASP2

To investigate the impact of phosphorylation at serine-835 on CAMSAP2's function in cell migration and Golgi reorientation, we developed MARK2 knockout HT1080 cells stably expressing both non-phosphorylatable (GFP-CAMSAP2 S835A) and phosphomimic (GFP-CAMSAP2 S835D) mutants (*Figure 5—figure supplement 1A*). In wound healing assays, the non-phosphorylatable S835A mutant exhibited a notable reduction in the ability to restore Golgi reorientation (*Figure 5A–C*). The results shown in *Figure 5A–C* indicate that in the absence of MARK2, there is no significant difference in Golgi reorientation between WT-CAMSAP2 and CAMSAP2 S835A. This observation supports the conclusion that MARK2-mediated phosphorylation of CAMSAP2 at S835 is essential for effective Golgi reorientation. The non-phosphorylatable S835A mutant resulted in a significant delay in wound

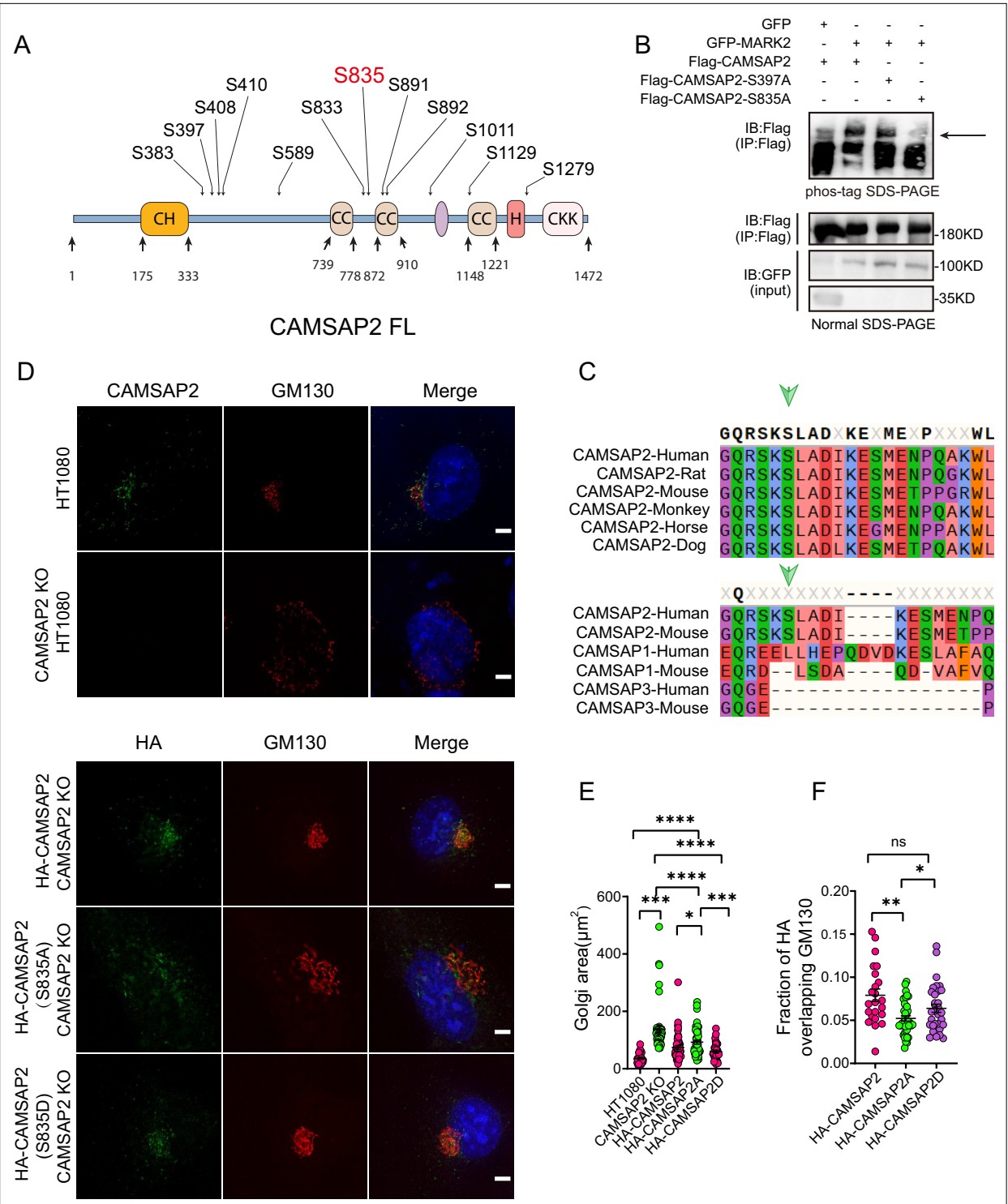

**Figure 4.** MARK2 phosphorylation of serine at position 835 of CAMSAP2 affects the localization of CAMSAP2 on the Golgi and regulates Golgi structure. (**A**) Mass spectrometry was used to detect the possible amino acid phosphorylation of CAMSAP2 by MARK2. The phosphorylation sites illustrated in (**A**) are derived from our analysis of the original mass spectrometry data. These sites were included based on their high confidence scores and data reliability. Importantly, only serine residues met the stringent criteria for inclusion, as no threonine or tyrosine residues had sufficient evidence for phosphorylation. (**B**) Proteins are expressed in HEK293 cells, and the band shift is examined using western blotting. The arrowhead labels the shifted bands. Confirmation of the site specificity of MARK2 phosphorylation. In order to better observe the upward shift of the bands, we compressed the image up and down. (**C**) The site phosphorylated by MARK2 is shown in different CAMSAP2 isoforms. Upper: Sequence alignment of CAMSAP2 orthologs across species. Bottom: Sequence alignment of CAMSAP2, CAMSAP1, and CAMSAP3 in mouse and human. Arrowheads indicate

*Figure 4 continued on next page*

*Figure 4 continued*

the CAMSAP2 S835. (**D**) In CAMSAP2 KO HT1080 cells, the expression of HA-CAMSAP2 (S835D) compensates for the CAMSAP2 KO effect on Golgi apparatus defects more effectively than HA-CAMSAP2 (S835A). Immunostaining of CAMSAP2 KO HT1080 cells expressing HA-CAMSAP2 (S835D) and HA-CAMSAP2 (S835A). Immunostaining of control cells with rat anti-HA (green) antibodies and mouse anti-GM130 antibodies (red). Nuclei were stained with DAPI (blue). Scale bar = 5 μm. (**E**) Area of the Golgi defined by the Golgi marker GM130 from (**D**). Each symbol represents an individual cell. Values = means ± SEM, unpaired two-tailed Student's *t*-test, ****$p<0.0001$, ***$p<0.001$, **$p<0.01$, *$p<0.05$ (N=3; HT1080: n=39; CAMSAP2 KO: n=50; CAMSAP2 KO (HA-CAMSAP2): n=50; CAMSAP2 KO (HA-CAMSAP2-S835A): n=49; CAMSAP2 KO (HA-CAMSAP2-S835D): n=35). (**F**) ImageJ software was used to calculate Manders' coefficient M1 values of HA-CAMSAP2 (S835A), HA-CAMSAP2 (S835D) co-localization with the Golgi marker GM130. Values = means ± SEM, unpaired two-tailed Student's *t*-test, *$p<0.05$ (N=3; CAMSAP2 KO (HA-CAMSAP2): n=23; CAMSAP2 KO (HA-CAMSAP2-S835A): n=40; CAMSAP2 KO (HA-CAMSAP2-S835D): n=32).

The online version of this article includes the following source data for figure 4:

**Source data 1.** PDF containing original scans of the relevant western blot analysis with highlighted bands and sample labels for *Figure 4B*.

**Source data 2.** Original file for the western blot analysis in *Figure 4B*.

**Source data 3.** Excel file containing the quantified data of statistic analysis for *Figure 4E and F*.

**Source data 4.** The mass spectrometry data supporting the identification of potential phosphorylation sites are provided for *Figure 4A*.

closure, in contrast to the phosphomimic S835D variant (*Figure 5—figure supplement 1B and C*). Previous studies have reported CAMSAP2's localization to the Golgi apparatus through interactions with proteins like CG-NAP (AKAP450) and CLASPs, with such protein interactions often modulated by phosphorylation (*Wu et al., 2016*). Our hypothesis was that phosphorylation alters CAMSAP2's interaction dynamics. However, immunoprecipitation analyses revealed no significant differences in the binding affinity between CAMSAP2 phosphomimic mutants and CG-NAP or CLASP2 (*Figure 5D–G*), implying that additional Golgi-associated proteins may play a role in CAMSAP2's localization during Golgi apparatus orientation.

## The phosphorylation of CAMSAP2 S835 promotes its interaction with the Golgi-associated protein USO1

Next, we employed TurboID proximity labeling technology combined with proteomic analysis (*Figure 6A*, *Figure 6—figure supplement 1A*; *Branon et al., 2018*). This methodology allowed us to identify Golgi-resident proteins that showed varying proximity interactions with CAMSAP2's phosphomimetic and non-phosphorylatable mutants (*Figure 6B*). Our selection criteria included two key points: first, the protein should demonstrate differential interaction with CAMSAP2 S835A/D in the mass spectrometry results, and second, it should be associated with Golgi structure. COPB1, SCAMP4, USO1, and DNM2 were chosen to validate the mass spectrometry results, among which USO1 demonstrated interaction with CAMSAP2 (*Figure 6—figure supplement 1B*). Our attention was particularly drawn to the Golgi-associated protein USO1, also known as p115, which binds GM130 and is localized to the membranes of cis-Golgi cisternae (*Brandon et al., 2003*; *Puthenveedu and Linstedt, 2004*; *Sohda et al., 2007*; *Satoh and Warren, 2008*; *Radulescu et al., 2011*; *Giacomello et al., 2019*). Further immunoprecipitation assay revealed that USO1 interacts with the region between the coiled-coil1 and CKK domains (*Figure 6—figure supplement 1C and D*). This region was previously reported to be important for the localization of CAMSAP2 in the Golgi apparatus (*Wu et al., 2016*; *Wang et al., 2017*). Importantly, our extended co-immunoprecipitation studies revealed marked differences in the interactions between USO1 and the two different forms of CAMSAP2, the phosphomimic (S835D) and the non-phosphorylatable (S835A) mutants (*Figure 6C and D*). These findings indicate that the phosphorylation status of serine-835 in CAMSAP2 plays a critical role in modulating its interaction with USO1.

Conversely, the knockdown of USO1 had a detrimental impact on the localization of CAMSAP2 within the Golgi apparatus (*Figure 6E–G*). Furthermore, overexpression experiments with USO1 in HT1080 cells demonstrated significant co-localization with CAMSAP2, highlighting the functional interplay between these two proteins (*Figure 6—figure supplement 2A and B*). To investigate whether the localization of USO1 on the Golgi changed during Golgi reorientation, we observed the co-localization of USO1 and GM130 by immunofluorescence staining at different time points of cell migration. Analysis revealed that the localization of USO1 on the Golgi was not affected during Golgi reorientation (*Figure 6—figure supplement 2C and D*). Finally, to comprehensively understand the

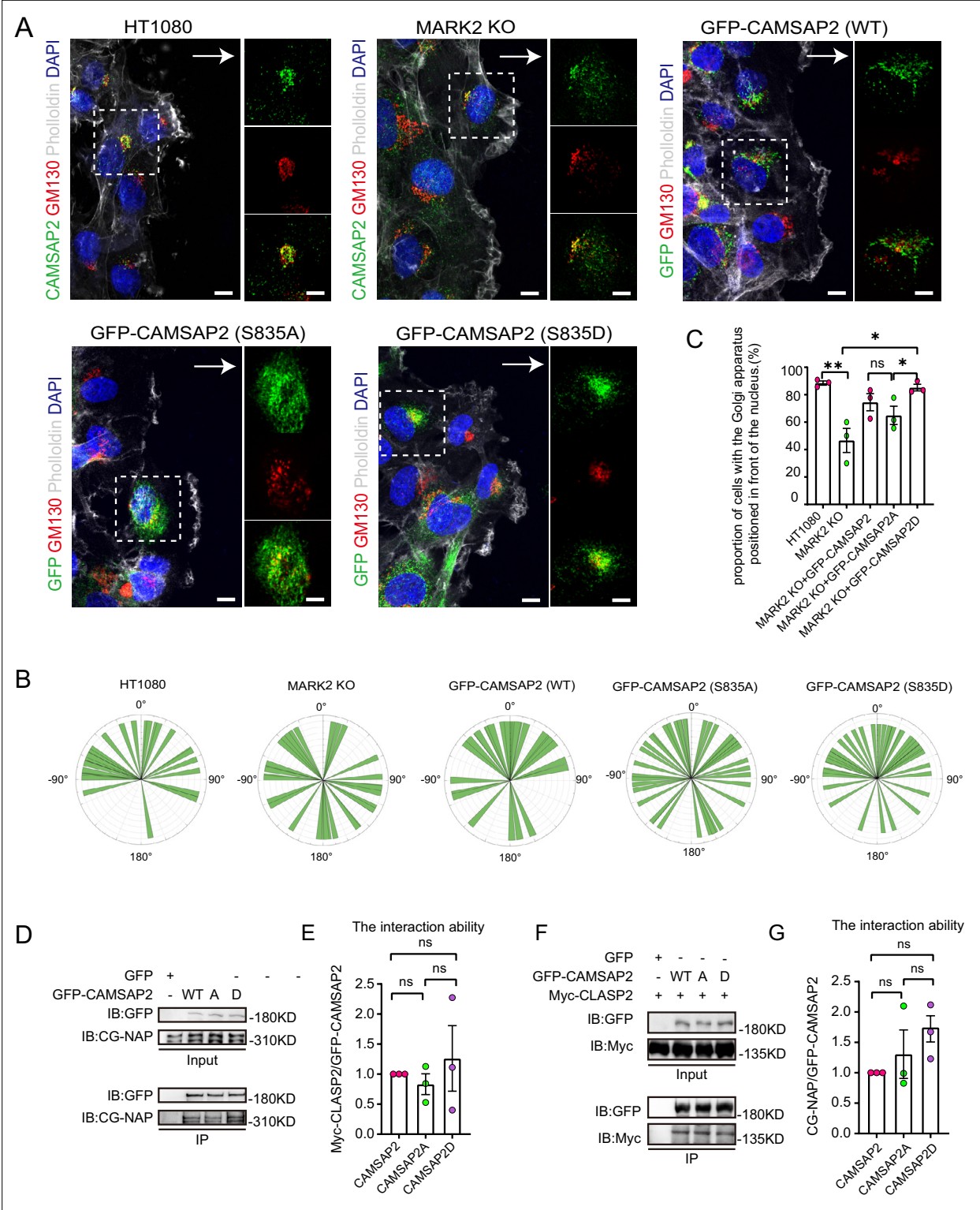

**Figure 5.** Identification of different interacting proteins with CAMSAP2 S835A/S835D. (**A**) Stable expression of CAMSAP2 S835A and CAMSAP2 S835D in MARK2 KO HT1080 was observed for Golgi reorientation at the cell wounding edge after 2 hr migration. The arrow points in the direction of the migration. On the right is a split-channel plot of the white dashed box on the left. We fixed the cells after 2 hr of migration with methanol for staining and assessed Golgi reorientation by measuring the angle between the Golgi centroid-nucleus centroid axis and the direction of cell migration. Black arrows point to the direction of migration. Scale bar = 5 μm. (**B**) Images demonstrate cells completing Golgi reorientation. If the absolute value of this angle is within 90 degrees, it indicates that the Golgi is positioned at the front edge of the nucleus and oriented in the migration direction, signifying

*Figure 5 continued on next page*

*Figure 5 continued*

proper Golgi reorientation. Conversely, if the absolute angle falls between 90 and 180 degrees, it suggests that the Golgi is positioned behind the nucleus, indicating incomplete Golgi reorientation (N=3; HT1080: n=34; MARK2 KO: n=32; GFP-CAMSAP2: n=28; GFP-CAMSAP2 S835D: n=40; and GFP-CAMSAP2 S835A: n=39). (**C**) Quantification of the proportion of cells in (**B**) that can complete Golgi reorientation properly. Values = means ± SEM, unpaired two-tailed Student's *t*-test (N=3; HT1080: n=34; MARK2 KO: n=32; GFP-CAMSAP2: n=28; GFP-CAMSAP2 S835D: n=40; and GFP-CAMSAP2 S835A: n=39). (**D**) Immunoblot showing the results of GFP-IP. There was no significant difference in the binding ability of CG-NAP to GFP-CAMSAP2, GFP-CAMSAP2 S835D, and GFP-CAMSAP2 S835A. The cells used in the experiment were 293T cells. CG-NAP (AKAP450) is indeed a 450 kDa protein, and the marker at 310 kDa represents the molecular weight marker's upper limit, above which CG-NAP is observed. (**E**) The quantitative results of the immunoblots presented in (**D**) are shown graphically. The signal intensity of CGNAP was normalized to the intensity of GFP in each mutant. Values = means ± SEM, unpaired two-tailed Student's *t*-test, p>0.05, N=3. (**F**) Immunoblot showing the results of GFP-IP. Myc-CLASP2 can bind to GFP-CAMSAP2, GFP-CAMSAP2 (S835D), and GFP-CAMSAP2 (S835A) with the same ability. The cells used in the experiment were 293T cells. (**G**) Quantitative results of the immunoblots shown in (**F**) are shown graphically. The signal intensity of Myc-CLASP2 was normalized to the intensity of GFP in each mutant. Values = means ± SEM, unpaired two-tailed Student's *t*-test, p>0.05, N=3.

The online version of this article includes the following source data and figure supplement(s) for figure 5:

**Source data 1.** PDF containing original scans of the relevant western blot analysis with highlighted bands and sample labels for *Figure 5D and F*.

**Source data 2.** Original file for the western blot analysis in *Figure 5D and F*.

**Source data 3.** Excel file containing the quantified data of statistic analysis for *Figure 5B, C, E, and G*.

**Figure supplement 1.** The phosphorylation of CAMSAP2 at S835 affects cell migration.

**Figure supplement 1—source data 1.** PDF containing original scans of the relevant western blot analysis with highlighted bands and sample labels for *Figure 5—figure supplement 1A*.

**Figure supplement 1—source data 2.** Original file for the western blot analysis in *Figure 5—figure supplement 1A*.

**Figure supplement 1—source data 3.** Excel file containing the quantified data of statistic analysis for *Figure 5—figure supplement 1C*.

dynamics between non-centrosomal microtubules and the Golgi apparatus during Golgi reorientation, we conducted cell wound-healing experiments (*Figure 6—figure supplement 2E and F*). Our observations revealed notable changes in the Golgi apparatus and microtubule network distribution in relation to the wounding. These findings corroborate our earlier results and suggest a highly dynamic interaction between the Golgi apparatus and microtubules during Golgi reorientation.

Directional cell migration refers to the coordinated movement of cells through cytoskeletal deformation, involving five sequential steps that occur in close spatial and temporal coordination. Initially, the cell responds to both intracellular and extracellular signals. Subsequently, cytosol extends lamellar pseudopods forward, while the Golgi apparatus undergoes transformation into a dispersed, unpolarized state, followed by the formation of a polarized Golgi body to complete Golgi reorientation, ultimately leading to tail retraction (*Figure 6H*, left). In this study, we have uncovered a novel polar signaling pathway that governs Golgi reorientation. MARK2 regulates Golgi reorientation by phosphorylating amino acid 835 of CAMSAP2, thereby modulating the binding affinity between CAMSAP2 and USO1 (*Figure 6H*, right).

## Discussion

The redirection of the Golgi apparatus during cell migration is a complex process involving multiple molecular interactions and the coordination of signaling networks. Our research has unveiled the central role of MARK2 in the phosphorylation of CAMSAP2 and its interaction with USO1 in Golgi apparatus reorientation. This mechanism underscores the importance of signaling pathways in cellular architectural adjustments, particularly during cell migration. The enrichment of phosphorylated CAMSAP2 at the Golgi apparatus may alter its spatial positioning by influencing microtubule stability and adjusting the composition of Golgi membrane proteins, thereby providing spatial guidance for directional migration. These findings not only offer a new perspective on the regulation of intracellular structures but also provide potential clues for understanding related diseases.

### The regulation of non-centrosomal microtubules in Golgi apparatus reorientation

The connection between the microtubule network and Golgi apparatus function is crucial, as evidenced by CAMSAP2's regulation of microtubule stability, which is essential for precise Golgi apparatus

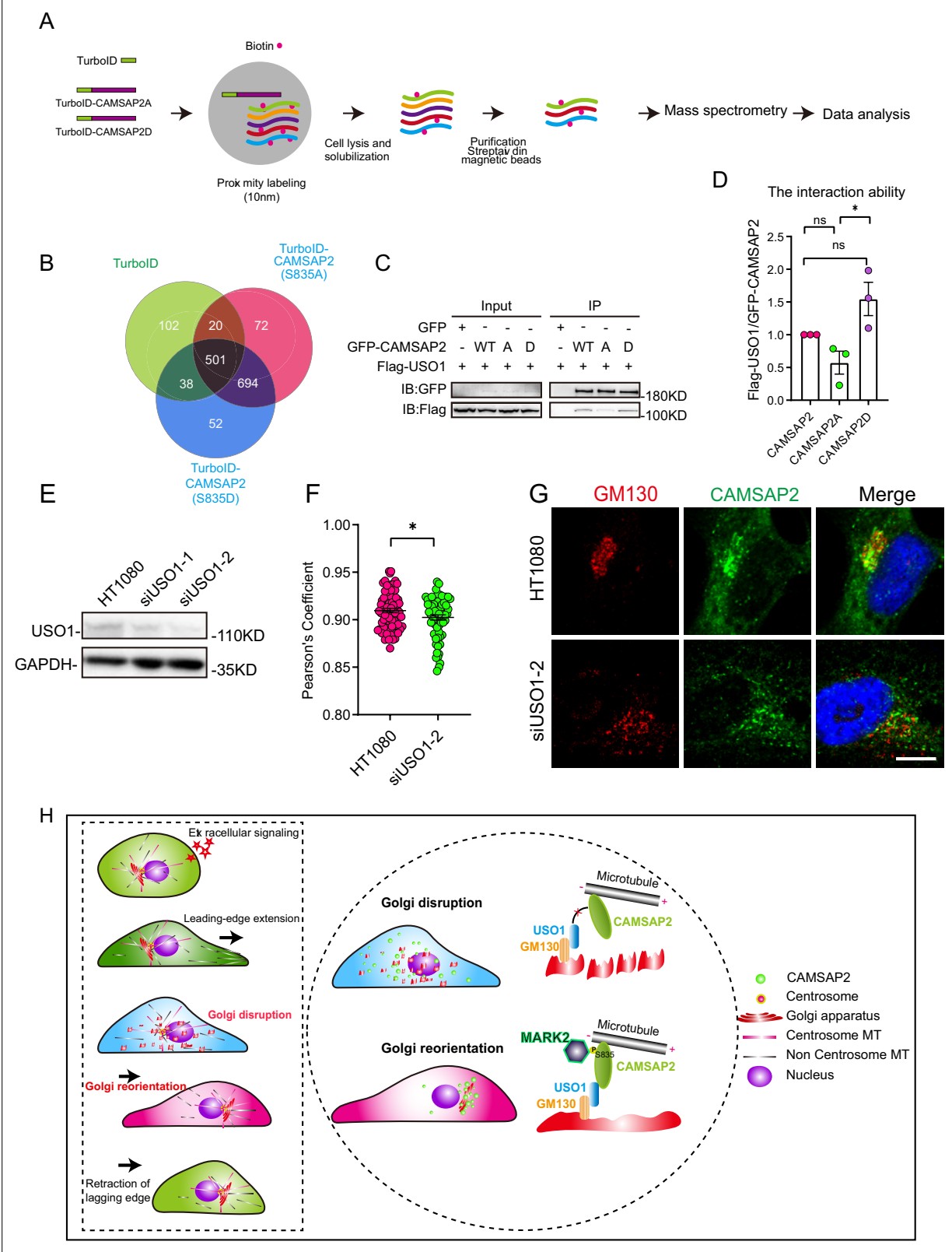

**Figure 6.** Differential interactions of USO1 with CAMSAP2 S835A/D. (**A**) Schematic overview of TurboID-mediated neighbor biotinylation assays using His-TurboID-CAMSAP2 (S835A), His-TurboID-CAMSAP2 (S835D), and His-TurboID. (**B**) Venn diagram of the three sets, generated using an online tool. Each colored circle represents a different dataset, and overlapping regions indicate common proteins. Proteins were identified that do not interact with His-TurboID but interact differently with FLAG-CAMSAP2 (S835A) and FLAG-CAMSAP2 (S835D). (**C**) Immunoblotting showing the results of GFP-IP.

*Figure 6 continued on next page*

*Figure 6 continued*

Flag-USO1 exhibits a stronger binding affinity with GFP-CAMSAP2 S835D than with FLAG-CAMSAP2 S835A. The cells used in the experiment were 293T cells. (**D**) Quantification of immunoblotting is shown in (**I**). The signal intensity of USO1 was normalized to the intensity of GFP in each mutant. Values = means ± SEM, unpaired two-tailed Student's *t*-test, *p<0.05, N=3. (**E**) Cell extracts were analyzed by immunoblotting using rabbit anti-USO1 and mouse anti-GAPDH to confirm the depletion of USO1. HT1080 cells were transfected with either control or USO1 siRNA for 72 hr. (**F**) Quantification of the localization of CAMSAP2 on the Golgi in different cells in (**E**). Each symbol represents an individual cell. Values = means ± SEM, unpaired two-tailed Student's *t*-test, *p<0.05 (N=3; control: n=77; USO1 siRNA: n=64). (**G**) Cells were stained with rabbit anti-CAMSAP2 (green) and mouse anti-GM130 (red) antibodies. Nuclei were stained with DAPI (blue). Scale bar = 5 μm. (**H**) The left rectangular dashed box represents the five steps of cell migration. The right circular dashed box responds to the regulatory mechanism that governs the second half of Golgi reorientation. The second half of Golgi reorientation: the process by which dispersed unpolarized Golgi stacks form a completely polarized Golgi.

The online version of this article includes the following source data and figure supplement(s) for figure 6:

**Source data 1.** PDF containing original scans of the relevant western blot analysis with highlighted bands and sample labels for *Figure 6C and E*.

**Source data 2.** Original file for the western blot analysis in *Figure 6C and E*.

**Source data 3.** Excel file containing the quantified data of statistic analysis for *Figure 6D and F*.

**Source data 4.** The mass spectrometry data for protein profiles identifying possible proteins are available in *Figure 6A*.

**Figure supplement 1.** USO1 interacts with CAMSAP2.

**Figure supplement 1—source data 1.** PDF containing original scans of the relevant western blot analysis with highlighted bands and sample labels for *Figure 6—figure supplement 1D, E, and F*.

**Figure supplement 1—source data 2.** Original file for the western blot analysis in *Figure 6—figure supplement 1D and E*.

**Figure supplement 2.** Significant changes in microtubules anchored to the Golgi apparatus during cell migration.

**Figure supplement 2—source data 1.** Excel file containing the quantified data of statistic analysis for *Figure 6—figure supplement 2B, D, and F*.

positioning. CAMSAP2 maintains microtubule stability by anchoring the minus end of microtubules to the Golgi apparatus, supporting both Golgi apparatus spatial localization and the establishment of cell polarity. The mechanism that we find reveals how the microtubule network, through molecules like CAMSAP2, is closely linked to Golgi apparatus function and subsequently influences cell migration direction and polarity establishment, emphasizing the importance of intracellular cytoskeletal organization in regulating cell behavior.

Additionally, signaling pathways play a vital role in the orientation of the Golgi apparatus. CG-NAP, as a multifunctional anchoring protein, can localize signaling molecules like protein kinase A (PKA) to specific cellular regions, such as the centrosome and Golgi apparatus (*Takahashi et al., 1999*; *Rivero et al., 2009*; *Kolobova et al., 2017*; *Ong et al., 2018*). By locally regulating PKA activity, CG-NAP participates in modulating microtubule stability and dynamics, thereby influencing Golgi apparatus positioning and cell polarity establishment (*Ong et al., 2018*). PKA activation can promote increased microtubule stability, leading to the anterior positioning of the Golgi apparatus, which supports directional cell migration (*Mavillard et al., 2010*; *Anacker et al., 2011*; *Chia et al., 2021*).

USO1 is a Golgi-associated protein closely associated with Golgi apparatus vesicle transport and membrane fusion processes (*Giacomello et al., 2019*). During cell migration, USO1 may influence Golgi apparatus structure and function by regulating interactions between the Golgi apparatus and vesicles, thereby participating in Golgi apparatus redirection (*Radulescu et al., 2011*). While the direct mechanisms of USO1's action require further investigation, it likely plays a crucial role in modulating the Golgi apparatus microenvironment, providing support for the proper positioning of the Golgi apparatus. We have observed that surface-associated proteins on the Golgi apparatus are selectively involved in its functional regulation.

The specific impact of CG-NAP and USO1 on Golgi apparatus repositioning during cell movement demonstrates how these two proteins jointly regulate the microtubule network associated with the Golgi apparatus. Through this regulation, they can influence cell migration capabilities, highlighting the direct impact of complex interactions between cellular structural elements on cell behavior and migration direction.

## Signaling transduction in Golgi apparatus orientation

Within the realm of cell migration, the precise spatial alignment of the Golgi apparatus is intricately linked to the dynamic reconfiguration of the microtubule network - a process meticulously governed by multiple signaling pathways that collaborate to orchestrate this intricate interplay.

A central protagonist in the Golgi apparatus's reorientation is MARK2, a protein kinase tasked with regulating the stability and polarity of non-centrosomal microtubules by phosphorylating target proteins, exemplified by CAMSAP2. Our research brings to the fore the specific role of MARK2 in the regulation of CAMSAP2, which concurrently oversees the microtubule network during cellular locomotion. Consequently, this regulatory oversight affects the interaction between CAMSAP2 and the Golgi-associated protein USO1. Notably, MARK2 facilitates the anchoring of the minus ends of non-centrosomal microtubules to the Golgi apparatus, thereby substantially bolstering the stability of the microtubule network. This phosphorylation-dependent regulation of CAMSAP2 by MARK2, along with its ensuing repercussions on the interplay between non-centrosomal microtubules and the Golgi apparatus, stands out as a pivotal mechanism underlying Golgi reorientation during the cellular migration journey. This regulatory mechanism ensures that the Golgi apparatus can adeptly respond to both intracellular and extracellular cues, thus enabling precise positioning along the trajectory of cell migration.

The localization of MARK2 at the Golgi was initially observed in experiments following serum starvation, where cells were fixed and stained (the data is not displayed). This observation was supported by the loss of Golgi localization in MARK2 knockdown cells, indicating the specificity of the antibody (the data is not displayed). However, this phenomenon was not consistently observed across all cells, likely due to its transient nature. We speculate that the localization of MARK2 to the Golgi depends on its activity and posttranslational modifications. For example, phosphorylation at T595 has been reported to regulate the translocation of MARK2 from the plasma membrane to the cytoplasm (*Hurov et al., 2004*). Serum starvation might induce modifications or conformational changes in MARK2, leading to its temporary Golgi localization. Additionally, we hypothesize that this localization may coincide with specific Golgi dynamics, such as the transition from dispersed to ribbon-like structures during cell migration. Exploring the reasons behind the localization of MARK2 at the Golgi apparatus in future research will be particularly intriguing.

Furthermore, several other players, including various signaling molecules and the cytoskeleton, partake in the intricate ballet of Golgi reorientation. RhoA, for instance, amplifies microtubule stability by catalyzing the action of microtubule-stabilizing proteins, deftly steering the Golgi apparatus toward the forefront of cell migration (*Glaven et al., 1999*; *Hall, 2005*; *Nalbant et al., 2009*; *Meiri et al., 2012*; *Eisler et al., 2018*; *Garcin and Straube, 2019*). Conversely, Rac1 and Cdc42 predominantly choreograph the assembly of actin filaments at the leading edge of cells (*Zhou et al., 2022*; *Hladyshau et al., 2023*). They achieve this by exerting influence over the growth and positioning of microtubules, thereby facilitating the Golgi's realignment toward the direction of cell migration. Moreover, the MAPK/ERK pathway champions the cause of microtubule growth and stability, thereby wielding influence over Golgi positioning and cellular polarity (*Sun et al., 2015*). The PI3K/Akt pathway, in turn, assumes the mantle of regulating the stability and dynamics of microtubules, thereby contributing to the establishment of cell polarity and the precise positioning of the Golgi apparatus (*Bucurica et al., 2023*). The mTOR signaling pathway, though indirectly, leaves its imprint on the microtubule network by modulating protein synthesis and cellular metabolism, thereby affecting microtubule stability and Golgi localization (*Makhoul et al., 2018*). Additionally, Wnt signaling lends its hand to the regulation of Golgi positioning by orchestrating the intricate dance of cell polarity and microtubule organization (*VanderVorst et al., 2019*). The combined synergy of these signaling pathways leaves an indelible mark on the precise Golgi positioning and the efficiency of cell migration by modulating the stability and dynamics of microtubules.

This collective body of evidence exemplifies how cells adeptly respond to both internal and external signals, leading to specific molecular events that recalibrate the spatial coordinates of the Golgi apparatus, consequently influencing the course of cell migration and cellular polarity. These discoveries underscore the pivotal role of signaling pathways in the governance of intracellular structural organization and dynamic adaptations, rendering invaluable insights into our comprehension of cell behavior. As we journey forward, further investigations will delve deeper into the specific mechanics governing these interactions, ultimately providing more profound insights into our understanding of cell migration.

Future research will delve deeper into the specific molecular interaction mechanisms among MARK2, CAMSAP2, and USO1, as well as how these interactions influence the dynamic repositioning of the Golgi apparatus during cell migration. Moreover, studies will focus on the biological significance

of these new findings, especially in processes such as cell migration, cell polarity establishment, and tissue development. Additionally, the role of these molecular mechanisms in diseases, particularly in cancer metastasis and other pathological conditions related to cell migration, will be a key focus of future research. Through this research, we hope to provide new targets for the treatment of related diseases and gain a deeper understanding of the regulation of Golgi apparatus reorientation during cell migration.

## Methods

### Cell culture and transfection

HT1080 and RPE1 cells were cultured in DMEM/F12 (Wisent) supplemented with 10% fetal calf serum (TransSerum) and 100 mg/ml penicillin and streptomycin at 37°C in 5% $CO_2$. HT1080 and RPE1 cells, kindly provided by Kangmin He, were confirmed by STR genotyping and verified to be mycoplasma-free using the Trans-Detect PCR Mycoplasma Detection Kit. MARK2 KO polyclonal cell lines were cultured in DMEM/F12 supplemented with 10% fetal bovine serum and 100 mg/ml penicillin and streptomycin at 37°C in 5% $CO_2$. CAMSAP2-KO polyclonal cell lines were cultured in DMEM/F12 supplemented with 10% fetal bovine serum and 100 mg/ml penicillin and streptomycin at 37°C in 5% $CO_2$. Cell lines stably expressing GFP-CAMSAP2, GFP-CAMSAP2-S835A, or GFP-CAMSAP2-S835D were cultured in DMEM/F12 supplemented with 10% fetal bovine serum and 100 mg/ml penicillin and streptomycin at 37°C in 5% $CO_2$. All cells were frozen in 10% DMSO (Alfa Aesar, cat. no. AAJ66650AP) in FBS at –80°C for a week, and then transferred to liquid nitrogen for preservation. Cells at approximately 50–60% confluence were transfected. Plasmids were transfected using Lipofectamine 2000 (Invitrogen) following the manufacturer's instructions. siRNAs were transiently transfected using Lipofectamine RNAiMAX transfection reagent (Invitrogen). The final concentration of siRNA duplexes was 10 nM. Cells transfected with plasmids were examined at 12–24 hr after transfection, and the effects of RNAi were examined at 72 hr.

**Table 1.** Information of primers.

| Construct | Primer | Sequence (5'–3') |
|---|---|---|
| | F | ATGACGGCGGCTGAGAAC |
| COPB1 | R | TTATATACTAGTTTTCTTCTGTGACAAGTTG |
| | F | CTGCGCTGGTTGGTGAAA |
| SCAMP4 | R | CAGGAATGAAGGGGAGGAGG |
| | F | ATGAATTTCCTCCGCGGGGTAATG |
| USO1 | R | CTAGATATGATCTAGATCCTTGCCAGG |
| | F | AGAGGAGCAAGgCCTTGGCAGAC |
| 835A | R | GTCTGCCAAGGcCTTGCTCCTCT |
| | F | CAGAGGAGCAAGgaCTTGGCAGACAT |
| 835D | R | ATGTCTGCCAAGtcCTTGCTCCTCTG |
| | F | CCGCAGACTCATgCTTCAGCCTCAG |
| 397A | R | CTGAGGCTGAAGcATGAGTCTGCGG |
| | F | TCAAGCTAAACCAGgCCAGTCCTGATAAC |
| 589A | R | GTTATCAGGACTGGcCTGGTTTAGCTTGA |
| | F | GAAAAGCTGAATTCAgCCTTGCACTTTCTAC |
| 892A | R | GTAGAAAGTGCAAGGcTGAATTCAGCTTTTC |
| | F | GTCTCTAGCCTCgCATTGGCATCG |
| 1279A | R | CGATGCCAATGcGAGGCTAGAGAC |

**Table 2.** Information of antibodies.

| Name | Species | Ordering company | Usage |
|---|---|---|---|
| GFP | MS | Proteintech (66002-1) | WB 1 : 4000 |
| GFP | Rb | MBL (589) | IF 1 : 500 |
| GFP | Chicken | Invitrogen (A10262) | IF 1 : 500 |
| Flag | Rb | Sigma (F7425) | IF 1 : 5000 |
| Flag | MS | Sigma (F3165) | WB 1 : 10,000 |
| HA | Rat | Roche (11867423001) | IF 1 : 250 |
| HA | MS | Sigma (H9658) | WB 1 : 20,000 |
| Myc | MS | Santa Cruz (SC-40) | IF 1 : 150 |
| Myc | Rb | MBL (562) | WB 1 : 2000 |
| GAPDH | MS | MBL (M171-3) | WB 1 : 2000 |
| α-Tubulin | Rb | ABclonal (AC025) | IF 1 : 50 |
| α-Tubulin | MS | Santa Cruz (SC-32293) | WB 1 : 1000 |
| α-Tubulin | MS | Sigma (T9026) | IF 1 : 500 |
| CAMSAP2 | Rb | Proteintech (17880-1-AP) | IF 1 : 1200 |
| MARK2 | Rb | abcam (ab136872) | IF 1 : 200 |
| MARK2 | Rb | abcam (ab133724) | WB 1 : 1000 |
| USO1 | Rb | Proteintech (13509-1-AP) | IF 1 : 2000, WB 1 : 2000 |
| CG-NAP | Rb | Laboratory-made antibodies | IF 1 : 1000, WB 1 : 2000 |
| GM130 | MS | BD (610823) | IF 1 : 800, WB 1 : 1500 |

Antibodies - polyclonal antisera against CG-NAP designated as αEE was prepared by immunizing rabbits with bacterially synthesized GST-fused fragments of aa 423–542.

## Plasmids

All cDNA expressed were cloned into pCMV-Tag2B to construct the Flag-tag plasmids. Flag-CAMSAP2 was described previously (*Ning et al., 2016*). Flag-CAMSAP2-S397A(Serine 397 to alanine substitution), Flag-CAMSAP2-S835 A (Serine 835 to alanine substitution), Flag-CAMSAP2-S589A (Serine 589 to alanine substitution), Flag-CAMSAP2-S892A (Serine 892 to alanine substitution), Flag-CAMSAP2-S1279A (Serine 1279 to alanine substitution) were modified from Flag-CAMSAP2, GFP-CAMSAP2, GST-MARK2, Flag-MARK2, Flag-MARK2-S, Flag-MARK2-ΔS, Flag-MARK2-ΔT, and Flag-MARK2-ΔST were described previously (*Zhou et al., 2020*). The cDNA of human COPB1, SCAMP4, USO1, and DNM2 was constructed into pCMV-Tag2B. Dr. Yuejia Li constructed GFP-CAMSAP2-740-1472, GFP-CAMSAP2-872-1472, GFP-CAMSAP2-1331-1472, GFP-CAMSAP2-1-739. pSpCas9 (BB)-2A-Puro (PX459) was a generous gift from Dr. Kangmin He. V5-TurboID-NES-pCDNA3 was a generous gift from Dr. Yingchun Wang. HA-CAMSAP2-S835A and HA-CAMSAP2-S835D were cloned into V5-TurboID-NES-pCDNA3. CAMSAP2 was cloned into the pOCC6_pOEM1-N-HIS6-EGFP vector expressed in Sf9 Purified His-GFP-CAMSAP2. The relevant primers are shown in *Table 1*.

## Antibodies

WB using secondary antibodies: Sheep HRP-conjugated anti-mouse IgG (NA931V, GE, 1: 7000). Donkey HRP-conjugated anti-rabbit IgG (NA934V, GE, 1: 10,000). IF using secondary antibodies: Goat anti-mouse, anti-rabbit, or anti-rat IgG, Alexa Fluor 488-, 555-, 647, IF 1: 500. Donkey anti-mouse, anti-rabbit, or anti-rat IgG, Alexa Fluor 488-, 555-, 647, IF 1: 500. First antibody was shown in *Table 2*.

## siRNAs

We used the following siRNAs purchased from Genepharma. Human MARK2 siRNA-1: 5′-CCT CCA GAA ACT ATT CCG CGA AGT ATT-3′. Human MARK2 siRNA-2: 5′-TCT TGG ATG CTG ATA TGA ACA TCA ATT-3′. Human USO1 siRNA-1: 5′-TCA TTA AGC AGC AGG AAA ATT-3′. Human USO1 siRNA-2: 5′-AAG ACC GGC AAT TGT AGT ACT TT-3′.

## CRISPR/Cas9 generation of KO clones

Stable polyclonal MARK2 and CAMSAP2 knockout cell lines were generated using CRISPR-Cas9 technology. CAMSAP2 KO and MARK2 KO HT1080 cells were generated by co-transfecting pX459-derived constructs. 24 hr post-transfection, co-transfected cells were selected in culture medium supplemented with 0.7 µg/ml puromycin (Invitrogen), until selection was complete. Efficiently transfected cells were isolated to generate clonal lineages by single-cell cloning in 96-well plates; colonies were all evaluated for KO efficiency by western blotting against the targeted protein. Only clones with 100% KO efficiency were conserved (*Pruett-Miller, 2015*; *Koch et al., 2018*).

> MARK2-KO Guide (CCAAGCTTCTCGGTTCCCTC)
> F1: ATGCGGCGGGTGCTCCTGCTGTG.
> R1: CCAGGTCAGGACTCGGAAGCGGGATGT
> CAMSAP2-KO Guide (CATGATCGATACCCTCATGA).
> F1: GGTCTTCTGTAGTTATTCTGCTGC.
> R1: CGTTTTCTGGCTATGAGATGCG

## Stable expression cell lines

For stable expression of GFP-CAMSAP2-S835A and GFP-CAMSAP2-S835D constructs in MARK2-KO cells, plasmids were transfected using Lipofectamine 2000 (Invitrogen) following the manufacturer's instructions, and 24 hr later, transfected cells were incubated in selection media in DMEM/F12 (Wisent) supplemented with 10% fetal calf serum and 0.6 mg/ml G-418. Cell death was monitored under the microscope until cell colonies started growing again, in approximately 7–14 days. Polyclonal cell lines expressing specific ranges of fluorescently tagged GFP-CAMSAP2-S835A and GFP-CAMSAP2-S835D were obtained by FACS gating and sorting for low levels of expression of the fluorescent protein, with untransfected HT1080 cells used as a control.

For stable expression of TurboID-CAMSAP2-S835A and TurboID-CAMSAP2-S835D constructs in HT1080 cells, plasmids were transfected using Lipofectamine 2000 (Invitrogen) following the manufacturer's instructions, and 24 hr later, transfected cells were incubated in selection media in DMEM/F12 (Wisent) supplemented with 10% fetal calf serum and 0.6 mg/ml G-418. Cell death was monitored under the microscope until cell colonies started growing again, in approximately 7–14 days. Afterward, we expanded a mixed stable expression cell line in selection media in DMEM/F12 (Wisent) supplemented with 10% fetal calf serum and 0.2 mg/ml G-418 for the next proximity protein labeling experiments.

## TurboID: proximity protein labeling

HT1080 cells stably expressing TurboID-CAMSAP2-S835A and TurboID-CAMSAP2-S835D or TurboID were treated with biotin (50 µM, Sigma-Aldrich) for 10 min before harvesting. Cell pellets were collected and lysed in ice-cold RIPA buffer (50 mM Tris-HCl [pH 7.5], 150 mM NaCl, 1% Triton X-100, 0.1% SDS, 0.5% sodium deoxycholate, supplemented with PhosSTOP and Protease inhibitor cocktail [MilliporeSigma, cat. no. P8849-5ML]), and lysates processed as before. Streptavidin enrichment was performed with Streptavidin Magnetic Beads (Thermo Fisher Scientific, cat. no. 88817) at 4°C for 12 hr. After affinity purification, wash the beads twice with RIPA lysis buffer (1 ml, 2 min at room temperature [RT]), once with 1 M KCl (1 ml, 2 min at RT), once with 0.1 M Na$_2$CO$_3$ (1 ml, ~10 s), once with 2 M urea in 10 mM Tris-HCl (pH 8.0) (1 ml, ~10 s) and twice with RIPA lysis buffer (1 ml per wash, 2 min at RT). Elute the enriched material from the beads by boiling each sample in 30 µl of 3× protein loading buffer supplemented with 2 mM biotin and 20 mM dithiothreitol (DTT) at 95°C for 10 min. Then, boil the samples at 95°C for 10 min (*Cho et al., 2020*). Streptavidin-horseradish peroxidase (HRP; Thermo Fisher Scientific, cat. no. S911). DTT (Thermo Fisher Scientific, cat. no. BP172-5). Protease inhibitor cocktail (MilliporeSigma, cat. no. P8849-5ML). BCA Protein Assay Kit (Thermo Fisher Scientific, cat.

no. 23225). BCA Protein Assay Kit (Thermo Fisher Scientific, cat. no. 23225). Biotin (Sigma-Aldrich, cat. no. B4501).

## In vitro kinase assay

*E. coli* containing the plasmid that encodes GST or GST-MARK2 were incubated at 37°C until A600 reached 0.6–0.8. To induce the expression of GST or GST-MARK2 proteins, isopropyl-β-D-thiogalactopyranoside was added to a final concentration of 0.1 mM. After being cultured for 18 hr at 20°C, cells were pelleted by centrifugation. Then, the bacteria were resuspended and sonicated for protein release. The proteins were purified with Glutathione-Sepharose 4B beads (GE Healthcare). His-GFP-CAMSAP2 was co-transfected with bacmids into Sf9 cells to generate the passage 1 (P1) virus; then, the P1 was added into Sf9 cells to generate the P2 virus. After confirming the expression of recombinant in P2 pellets via western blotting, P2 virus was added into Sf9 for protein expression. Expression was completed for 72 hr, and the Sf9 cells were collected. The pellets were resuspended in HTK buffer (25 mM HEPES pH 8.0, 150 mM KCl, 1 mM tris(2-carboxyethyl) phosphine, 10% [wt/vol] glycerol, 0.25 M NaCl, 1 mM DTT, 0.1% Triton X-100) and sonicated to homogenize samples; proteins fused with His-tag in the supernatant were purified by using NiNTA beads according to the manufacturer's instructions. All purified proteins were snap-frozen in liquid nitrogen and stored at –80°C. GST or GST-MARK2 was incubated with GFP-CAMSAP2 in kinase buffer (50 mM Tris-HCl pH 7.5, 12.5 mM $MgCl_2$, 1 mM DTT, 400 μM ATP) at 30°C for 30 min. Samples were boiled with loading buffer to stop the reaction (*Han et al., 2020*; *Zhou et al., 2020*; *Xue et al., 2024*).

## Immunofluorescence

For most of the immunostaining experiments, cells were fixed with ice-cold methanol for 5 min at –20°C, washed three times with PBST (0.1% Triton X-100 in PBS) for 5 min and incubated for 30 min with 1% BSA at RT. Cells were incubated with primary antibodies diluted by blocking buffer for 1–2 hr and washed three times with PBST (0.1% Triton X-100 in PBS) for 5 min. Cells were incubated with secondary antibodies for 1–2 hr. Coverslips were mounted in FluorSave reagent (MilliporeSigma). For MARK2 immunofluorescence staining, before fixing the cells, the serum in the culture medium should be reduced to 1% and cultured for 2 hr.

## Western blot

Cells were lysed in RIPA buffer (50 mM Tris-HCl, 150 mM NaCl, 1 mM ethylenediamine tetraacetic acid [EDTA], 1% NP-40, 0.5% DOC, 0.1% SDS, pH 8.0) containing 1× protease inhibitor cocktail. For phospho-experiments, cells were lysed in a phospho-buffer (20 mM Tris-HCl pH 8.0, 150 mM NaCl, 1 mM EDTA, 1 mM $Na_3VO_4$, 25 mM NaF, 10 mM $Na_2P_2O_7$, 50 mM β-glycerophosphate, 1% NP-40), containing protease inhibitor cocktail and phosphatase inhibitor cocktail. After centrifuging to remove insoluble fractions, sample loading was added and boiled for 5 min. Protein was separated on SDS-PAGE gels (to determine the band shift of MARK2-dependent phosphorylation, 30 μM Phos-tag reagent was mixed in gels following the manufacturer's instructions) and transferred to PVDF membranes. The membranes were blocked for 1 hr in 5% BSA, then incubated with primary antibody at 4°C overnight. After washing three times with TBST, membranes were incubated with HRP-conjugated secondary antibody and visualized with enhanced chemiluminescence. Insoluble fractions and sample loading were added and boiled for 5 min. For immunoprecipitation, cultured cells were lysed in cold lysis buffer (20 mM Tris-HCl pH 8.0, 150 mM NaCl, 1 mM EDTA, 1% NP-40), containing a protease inhibitor cocktail. After centrifuging to remove the insoluble fraction, GFP-beads were added into the supernatant at 4°C for 1.5 hr. The beads were subsequently washed three times in the wash buffer (20 mM Tris-HCl, 150 mM NaCl, 1 mM EDTA, 0.1% NP-40, pH 8.0).

## Isolation of Golgi-enriched fractions

Cells from a 4×14 $cm^2$ culture dish were trypsinized and harvested at 500×*g* for 10 min and washed twice in cold PBS with centrifugation at 500×*g* for 10 min. The pellet was washed with cold homogenization buffer, and the cells were spun down at 500×*g* for 10 min. The pellet was resuspended in 5 volumes of cold homogenization buffer followed by homogenization using a homogenizer. Post-nuclear supernatant (PNS) was obtained after centrifugation of the cell homogenate at 600×*g* for 10 min at 4°C. To 12 ml of PNS, 11 ml of 62% (wt/wt) sucrose solution and 250 ml of 100 mM EDTA

(pH 7.1) were added to obtain a homogenate with 37% (wt/wt) sucrose concentration. Four milliliters of this homogenate were placed into an SW40 tube Beckman and overlaid with a 5 ml 35% (wt/wt) and a 4 ml 29% (wt/wt) layer of sucrose solution (in 10 mM Tris, pH 7.4). These gradients were centrifuged for 2 hr and 40 min at 100,000×$g$. Approximately 1 ml of a Golgi-enriched fraction was collected at the interphase of 35–29% sucrose layers. For further analysis, the collected membranes were pelleted by centrifugation for 30 min at 100,000×$g$ at 4°C after the addition of 4 volumes of PBS to 1 volume of the Golgi-enriched fraction. Samples were boiled with a loading buffer. Alternatively, the collected membranes are stored at –80°C for future experiments (*Song et al., 2019*).

### Wound healing and migration

A confluent HT1080 monolayer was scratched using a yellow pipette tip to create a cell-free zone. Fields were photographed just after injury and 2 hr or 8 hr later. Quantification of cell migration was made by measuring the percentage of area recovery using ImageJ software. Alternatively, phase-contrast imaging was performed (*Rodriguez et al., 2005*).

### Mass spectrometry

To determine the MARK2-dependent phosphorylation sites in CAMSAP2, GST or GST-MARK2 was incubated with GFP-CAMSAP2 in kinase buffer (50 mM Tris-HCl pH 7.5, 12.5 mM MgCl$_2$, 1 mM DTT, 400 µM ATP) at 30°C for 30 min. Samples were boiled with a loading buffer to stop the reaction. After being verified by western blotting (Phos-tag was added into the SDS-PAGE gels), the sample was loaded onto SDS-PAGE gels and stained with CBB. The bands of GFP-CAMSAP2 were cut out and delivered to mass spectrometry analysis.

### Quantification of western blotting

Quantification of protein levels was performed with ImageJ. The bands representing respective proteins were manually outlined, and the signal intensity was measured. For normalized protein levels, α-tubulin or GAPDH intensity in each band was also measured with ImageJ. The graphs of the intensity display the values relative to the maximum value (100%) in each experiment.

### Microscopy image acquisition, processing, and quantitative analysis

Images were obtained through Zeiss Observer Z1. For superresolution confocal imaging of immunostained samples, Zeiss LSM980 with Airyscan 2 superresolution system was performed with a ×63 1.4 NA objective. All images were modified by adjustments of levels and contrast. To improve image resolution and contrast, sparse deconvolution was used by following the previous report (*Zhao et al., 2021*). Images from western blotting of Phos-tag (Wako) have modified the length in the Y-axis to adjust the bands, as the protein bands in Phos-tag SDS-PAGE become broad and tailing. Cells were carefully stained and recorded under the same conditions for quantification of co-localization. Pearson's R value above the threshold was calculated with the Coloc 2 plug-in in Fiji software from 8-bit 512×512 pixel images of the region of interest with low background.

### Data statistics

The results of the experiments in this study were obtained from three or more independent replications. GO analysis of proteins was plotted by https://www.bioinformatics.com.cn, an online platform for data analysis and visualization. The results of quantitative analyses were expressed as 'mean ± SEM'. Immunofluorescence staining images were analyzed using ImageJ software to count the fluorescence intensity. Data were analyzed using a two-tailed Student's $t$-test. Differences were not significant (NS) when $p > 0.05$, significant (*) when $p < 0.05$, highly significant (**) when $p < 0.01$, and highly significant (***) when $p < 0.001$. All data were statistically analyzed using GraphPad Prism 6.0 software.

## Acknowledgements

We thank Dr. Kangmin He. HT1080 and RPE1 cells were generous gifts from Dr. Kangmin He. We thank Jinhui Shao for his help in gene editing. We thank Guangzhou CSR Biotech Co. Ltd for providing sparse deconvolution software MicroscopeX FINER to improve image quality and valuable discussion. We thank Xing Jia and Qing Bian for their technical support with LSM980 Airyscan imaging and image

analysis. This work was funded by the National Key Research and Development Program of China (2021YFA0804802) and the National Natural Science Foundation of China (31930025 and 32100760).

## Additional information

### Funding

| Funder | Grant reference number | Author |
| --- | --- | --- |
| National Key Research and Development Program of China | 2021YFA0804802 | Wenxiang Meng |
| National Natural Science Foundation of China | 31930025 | Wenxiang Meng |
| National Natural Science Foundation of China | 32100760 | Zhengrong Zhou |

The funders had no role in study design, data collection and interpretation, or the decision to submit the work for publication.

### Author contributions

Peipei Xu, Ruifan Lin, Data curation, Formal analysis, Methodology; Rui Zhang, Data curation, Formal analysis, Investigation, Methodology; Zhengrong Zhou, Yuejia Li, Formal analysis, Funding acquisition, Methodology; Honglin Xu, Data curation; Mengge Yang, Data curation, Funding acquisition, Investigation; Yingchun Wang, Investigation, Project administration; Xiahe Huang, Methodology; Qi Xie, Supervision, Investigation; Wenxiang Meng, Conceptualization, Funding acquisition, Investigation, Writing – review and editing

### Author ORCIDs

Peipei Xu ⓘ https://orcid.org/0009-0001-6270-1713
Honglin Xu ⓘ https://orcid.org/0000-0003-4554-862X
Wenxiang Meng ⓘ https://orcid.org/0000-0003-0060-0535

Reviewer #1 (Public review): https://doi.org/10.7554/eLife.105977.2.sa1
Reviewer #2 (Public review): https://doi.org/10.7554/eLife.105977.2.sa2
Author response https://doi.org/10.7554/eLife.105977.2.sa3

## Additional files

### Supplementary files

MDAR checklist

### Data availability

All data generated or analysed during this study are included in the manuscript and supporting files.

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
