## [Editor Report · eLife Assessment]

The authors propose that the kinase MARK2 regulates the Golgi's reorientation towards the cell's leading edge through the regulation of microtubule binding protein CAMSAP2 and its binding to USO1. While the model is interesting and the study is **useful**, the quantification of an insufficient number of cells and insufficient description of the methods and biological replicates mean the results are **inadequate** to support the model.

[Editors' note: this paper was reviewed by Review Commons.]

---

## [Referee Report · Reviewer #1 (Public review)]

Summary:

This work by the Meng lab investigates the role of the proteins MARK2 and CAMSAP2 in the Golgi reorientation during cell polarisation and migration. They identified that both proteins interact together and that MARK2 phosphorylates CAMSAP2 on the residue S835. They show that the phosphorylation affects the localisation of CAMSAP2 at the Golgi apparatus and in turn influences the Golgi structure itself. Using the TurboID experimental approach, the author identified the USO1 protein as a protein that binds differentially to CAMSAP2 when it is itself phosphorylated at residue 835. Dissecting the molecular mechanisms controlling Golgi polarisation during cell migration is a highly complex but fundamental issue in cell biology and the author may have identified one important key step in this process.

Comments on latest version:

I thank the authors for the numerous revisions they have made to this manuscript, which have strengthened its clarity and overall quality. However, I must reiterate my initial concerns from the first review regarding the rigor of the data analysis, as certain methodological choices may lead to potential overinterpretation of the results.

For instance, the low number of cells analyzed in the new Figure 1B (N = 3; 0 h: n = 28; 0.5 h: n = 23; 2 h: n = 20) indicates that fewer than 10 fixed cells have been quantified per replicate. Given the variability of the CAMSAP2 signal observed in Supplementary Figure 2, this sample size does not appear optimal for accurately capturing the complexity of CAMSAP2 localization within the cell population. Additionally, the Pearson's coefficients calculated between CAMSAP2 and GM130 in Figure 1B (approximately 0.4) do not align with those in Figure 3C, where the control condition shows values above 0.6. This discrepancy highlights the high variability of CAMSAP2 Golgi localization in the HT1080 cell population, which may not be adequately represented by the quantification of such a limited number of cells. If the population distribution were narrow, averaging only a few cells might be sufficient to achieve high statistical power; however, this does not appear to be the case, and a larger sample size is necessary.

Furthermore, to ensure a more robust analysis, SuperPlots displaying each biological replicate should be provided for all quantifications, and statistical analysis should be conducted using a t-test or ANOVA on the means of the three independent experiments rather than on the total number of cells, as the latter approach may influence statistical significance (for reference: jcb.202001064). This recommendation is relevant for Figures 1E, 3B, 3C, 4E, 4F, 6F, Sup1D, Sup3D, Sup3E, Sup3I, and Sup3G and should be implemented whenever possible.

For instance, in the new Figure 6F, the statistical difference (1 star) between Pearson's coefficients for HT1080 and siUSO1-2 conditions, both approaching 90, raises questions about whether this difference is truly substantial enough to support the claim that USO1 knockdown negatively impacts CAMSAP2 localization.

Publishing in journals such as eLife requires high standards in data analysis to ensure rigorous and reproducible scientific conclusions. In its present form, this manuscript does not yet meet those standards.

Additional comments:

Supplementary figure 2

A) The video microscopy conditions are not described in the Materials and Methods section. It is unclear what type of microscope was used-was it a bright-field or confocal microscope? The images contain a significant amount of out-of-focus signal, making it difficult to accurately assess the extent of Golgi apparatus dispersion as described by the authors. If a confocal microscope was used, a Z-stack projection would be beneficial for quantifying this process.

---

## [Referee Report · Reviewer #2 (Public review)]

Summary:

The manuscript Xu et al. explores the regulation of the microtubule minus end protein CAMSAP2 localization to the Golgi by the Serine/threonine-protein kinase MARK2 (PAR1, PAR1B). The authors show that depletion of MARK2 alters Golgi morphology and diminishes CAMSAP2 localization to the Golgi apparatus. The authors combine mass spectroscopy and immunoprecipitation to show that CAMSAP2 is phosphorylated at S835 by MARK2, and that this phosphorylation regulates localization of CAMSAP2 at Golgi membranes. Further, the authors identify USO1 (p115) as the Golgi resident protein mediating CAMSAP2 recruitment to the Golgi apparatus following S835 phosphorylation.

Impact:

The Golgi apparatus is a key organelle in cell migration- post translationally modifying and sorting cargo for directed trafficking, acting as a signalling hub, whilst functioning as a nucleation site for microtubules. These functions are essential to establish cell polarity during migration, highlighting the importance of understanding how cells reorient their Golgi in response to different environmental cues.

The study will be of interest to fundamental biologists investigating Golgi function, and positioning, particularly in the context of different cell migration settings. It may also interest scientists investigating the loss of polarity in cancer or the maintenance of epithelial tissue architecture. I am a cell biologist with expertise in cell trafficking, cytoskeleton, and cell migration- during processes spanning development, homeostasis and cancer.

Comments on latest version:

Labelling of graphs - many of the graphs are comparing HT1080 cells with two conditions: parental and KO i.e. Figure 2F, H, I. The labels the authors use are "HT1080 and CAMSAP2 KO". This is confusing and should be changed to "parental and CAMSAP2 KO", the cell type HT1080 could be listed in the figure legend or on the graph below the conditions. (Similar to the labelling in Figure 3 H, I where they use "control and siRNA").

The method section needs improvement - particularly around analysis methods, and statistical details for experiments. I recommend a supplementary table outlining exactly where the data is from (pooled, biological/technical repeats, n definitions, tests of normality etc).

---

## [Author Response]

**Reviewer #1:**
The manuscript Xu et al. explores the regulation of the microtubule minus end protein CAMSAP2 localization to the Golgi by the Serine/threonine-protein kinase MARK2 (PAR1, PAR1B). The authors utilize immunofluorescence and biochemical approaches to demonstrate that MARK2 is localized at the Golgi apparatus via its spacer domain. They show that depletion of this protein alters Golgi morphology and diminishes CAMSAP2 localization to the Golgi apparatus. The authors combine mass spectroscopy and immunoprecipitation to show that CAMSAP2 is phosphorylated at S835 by MARK2, and that this phosphorylation regulates localization of CAMSAP2 at Golgi membranes. Further, the authors identify USO1 (p115) as the Golgi resident protein mediating CAMSAP2 recruitment to the Golgi apparatus following S835 phosphorylation. The authors would need to address the following queries to support their conclusions.

We sincerely thank the reviewer for their valuable time and effort in evaluating our manuscript. We deeply appreciate the constructive feedback and insightful suggestions, which have been instrumental in improving the quality and clarity of our study. We have carefully considered all the comments and have made the necessary revisions to address the concerns raised.

Major Comments(1) Dynamic localization of CAMSAP2 during Golgi reorientation- The authors use fixed wound edges assays and co-localization analysis to describe changes in CAMSAP2 positioning during Golgi reorientation in response to polarizing cues (a free wound edge in this case). In Figure 1C, they present a graphical representation of quantified immunofluorescence images, using color coding to to describe the three states of Golgi reorientation in response to a wound (green, blue, red indicating non-polarised, partial and complete Golgi reorientation, respectively). They then use these 'colour coded' classifications to quantitate CAMSAP2/GM130 co-localization.It is unclear why the authors have not just used representative immunofluorescence images in the main figures. Transparent, color overlays could be placed over the cells in the representative images to indicate which of the three described states each cell is currently exhibiting. However, for clarity, I would recommend changing the color coded 'states' to a descriptor rather than a color. i.e. Figure 1D x axis labels should be 'complete' and 'partial', instead of 'red' and 'blue'.

Thank you for this insightful suggestion. We have added representative immunofluorescence images with transparent color overlay to indicate the three Golgi orientation states. These images are included in Supplementary Figure 2B-C, providing a clear visual reference for the quantitative data. Additionally, we have revised the x-axis labels in Figure 1E from "Red" and "Blue" to "Complete" and "Partial" to ensure clarity and consistency with the descriptive terminology in the text. These changes are described in the Results section (page 7, lines 15-19) and the figure legend (page 29, lines 27-29).

We believe these updates improve the clarity and accessibility of our figures and hope they address the reviewer’s concerns.

- note- figure 2 F-G, is semi quantitative, why did the authors not just measure Golgi angle using the nucleus and Golgi distribution?

We appreciate the reviewer’s comment on this point. Following the recommendation, we have performed an additional analysis measuring Golgi orientation angles based on the nucleus-Golgi distribution. This quantitative approach complements our initial semi-quantitative analysis and provides a more precise assessment of Golgi orientation during cell migration.

The new data have been incorporated into Supplementary Figure 1F-H. These results clearly demonstrate the consistency between the quantitative and semi-quantitative methods, further validating our findings and highlighting the dynamic changes in Golgi orientation during cell migration. These changes are described in the Results section (page 6, lines 24-31).

- While it is established that the Golgi is dispersed during reorientation in wound edge migration, the Golgi apparatus also becomes dispersed/less condensed prior to cell division. As the authors have used fixed images - how are they sure that the Golgi morphology or CAMSAP2 localization in 'blue cells' are indicative of Golgi reorientation and not division? Live imaging of cells expressing CAMSAP2, and an additional Golgi marker could be used to demonstrate that the described changes in Golgi morphology and CAMSAP2 localization are occurring during the rear-to-front transition of the Golgi.

Thank you for raising this important question. To address this concern, we carefully examined the nuclear morphology of dispersed Golgi cells and found no evidence of mitotic features, indicating that these cells are not undergoing division (Figure 1A, Supplemental Figure 2A). Furthermore, during the scratch wound assay, we use 2% serum to culture the cells, which helps minimize the impact of cell division. This analysis has been added to the Results section (page7, lines 19-22 in the revised manuscript).

Additionally, we conducted live-cell imaging, as suggested, using cells expressing a Golgi marker. This approach confirmed that Golgi dispersion occurs transiently during reorientation in cell migration. The new live-cell imaging data have been incorporated into Supplementary Figure 2A, and the corresponding description has been updated in the Results section (page 7, lines 2-5).

Finally, considering that overexpression of CAMSAP2 can lead to artifactually condensed Golgi structures, we used endogenous staining to observe CAMSAP2 localization at different stages of migration. These observations provide a clearer understanding of CAMSAP2 dynamics during Golgi reorientation and are now presented in revised Figure 1A-B. This information has been described in the Results section (page 7, lines 5-10).

We hope these additions and clarifications address the reviewer’s concerns. Once again, we are deeply grateful for this constructive feedback, which has greatly improved the robustness of our study.

(2) MARK2 localization to the Golgi apparatus- The authors investigated the positioning of endogenous MARK2 via immunofluorescence staining, and exogenous flag-tagged MARK2 in a KO background. The description of the protocol required to visualize Golgi localization of MARK2 is inconsistent between the results and methods text. The results text reads as through the 2% serum incubation occurs as a blocking step following fixation. Conversely, the methods section describes the 2% serum incubation as occurring just prior to fixation as a form of serum starvation. The authors need to clarify which of these protocols is correct. Further, whilst I can appreciate that the mechanistic understanding of why serum starvation is required for MARK2 Golgi localization is beyond the scope of the current work, the authors should at a minimum speculate in the discussion as to why they think it might occur.

We sincerely thank the reviewer for the constructive feedback on the localization of MARK2 at the Golgi. Due to the complexity and variability of this phenomenon, we decided to remove the related data from the current manuscript to maintain the rigor of our study. However, we have included a discussion of this phenomenon in the Discussion section (page 13, lines 31-39 and page 14, 1-6in the revised manuscript) and plan to further investigate it in future studies.

The localization of MARK2 at the Golgi was initially observed in experiments following serum starvation, where cells were fixed and stained (The data is not displayed). This observation was supported by the loss of Golgi localization in MARK2 knockdown cells, indicating the specificity of the antibody (The data is not displayed). However, this phenomenon was not consistently observed across all cells, likely due to its transient nature.We speculate that the localization of MARK2 to the Golgi depends on its activity and post-translational modifications. For example, phosphorylation at T595 has been reported to regulate the translocation of MARK2 from the plasma membrane to the cytoplasm (Hurov et al., 2004). Serum starvation might induce modifications or conformational changes in MARK2, leading to its temporary Golgi localization. Additionally, we hypothesize that this localization may coincide with specific Golgi dynamics, such as the transition from dispersed to ribbon-like structures during cell migration.

We also acknowledge the inconsistency in the Results and Methods sections regarding serum starvation. We confirm that serum starvation was performed prior to fixation as an experimental condition, rather than as a blocking step in immunostaining. This clarification has been incorporated into the revised Methods section (page 24, lines 11-12).

We hope this clarification, along with our planned future studies, adequately addresses the reviewer’s concerns. Once again, we deeply appreciate the reviewer’s valuable comments, which have provided important insights for our ongoing work. References：

Hurov, J.B., Watkins, J.L., and Piwnica-Worms, H. (2004). Atypical PKC phosphorylates PAR-1 kinases to regulate localization and activity. Curr Biol 14 (8): 736-741.

- The authors should strengthen their findings by using validated tools/methods consistent with previous publications. i.e. Waterman lab has published two MARK2 constructs- Apple and eGFP tagged versions (doi.org/10.1016/j.cub.2022.04.088), and the localization of MARK2 in U2Os cells using the same antibody (Anti- MARK2 C-terminal, ABCAM Cat# ab136872). The authors should (1) image the cells live using eGFP-tagged MARK2 during serum starvation to show the dynamics of this localization, (2) image U2Os cells using the abcam ab136872 antibody +/- 2% serum starve. Two MARK2 antibodies are listed in Table 2. Does abcam (ab133724) show a similar localisation?- The Golgi localization of MARK2 occurs in the absence of the T structural domain, but not when full length MARK2 is expressed. The authors conclude the T- domain is likely inhibitory. When combined with the requirement for serum starvation for this interaction to occur, the authors should clarify the physiological relevance of these observations.

We sincerely thank the reviewer for their valuable suggestions regarding the use of tools and methods and the physiological relevance of MARK2 localization to the Golgi. Regarding the question of how MARK2 itself localizes to the Golgi, we are currently unable to fully elucidate the underlying mechanism. Therefore, we have removed the discussion of MARK2’s Golgi localization from the manuscript to ensure scientific accuracy. However, Below, we provide our detailed response as soon as possible:

First, regarding the suggestion to use tools and methods developed by the Waterman lab to strengthen our findings, we have carefully evaluated their applicability. In our live-cell imaging experiments, we found that full-length MARK2 does not stably localize to the Golgi, even under serum starvation conditions. However, truncated MARK2 mutants lacking the Tail (T) domain exhibit robust Golgi localization. Furthermore, our immunofluorescence staining results indicate that the Spacer domain is the minimal region required for MARK2 localization at the Golgi. Based on these findings, we believe that live-cell imaging of EGFP-tagged full-length MARK2 may not effectively reveal the dynamics of its Golgi localization. However, we plan to focus on the truncated constructs in future studies to better explore the mechanisms underlying MARK2's dynamic behavior.

Regarding the use of the ab136872 antibody to stain U2OS cells with and without serum starvation, we note that the protocol described by the Waterman lab involves pre-fixation and permeabilization steps, which are not compatible with live-cell imaging. Additionally, we observed that MARK2 Golgi localization appears to be condition-dependent and may coincide with specific Golgi dynamics, such as transitions from dispersed stacks to intact ribbon structures. These events are likely brief and challenging to capture consistently. Nevertheless, we recognize the value of this experimental design and plan to adapt the staining conditions in future work to validate our results further. As for the ab133724 antibody listed in Table 2, we clarify that it has only been validated for Western blotting in our study and does not yield reliable results in immunofluorescence experiments. For this reason, all immunofluorescence staining in this study relied exclusively on ab136872. This distinction has been clarified in the revised Table 2 .

Regarding the hypothesis that the Tail domain of MARK2 is inhibitory, our observations showed that truncated MARK2 mutants lacking the T domain stably localized to the Golgi, whereas fulllength MARK2 did not. Literature evidence supports this hypothesis, as studies on the yeast homolog Kin2 indicate that the C-terminal region (including the Tail domain) binds to the Nterminal catalytic domain to inhibit kinase activity (Elbert et al., 2005). We speculate that serum starvation disrupts this intramolecular interaction, relieving the inhibition by the T domain, activating MARK2, and promoting its localization to the Golgi. Moreover, we hypothesize that the transient nature of MARK2 localization to the Golgi may be related to specific Golgi remodeling processes, such as the transition from dispersed stacks to intact ribbon structures during cell migration or polarity establishment.

References：

Elbert, M., Rossi, G., and Brennwald, P. (2005). The yeast par-1 homologs kin1 and kin2 show genetic and physical interactions with components of the exocytic machinery. Mol Biol Cell 16 (2): 532-549.

(3) Phosphorylation of CAMSAP2 by MARK2- The authors examined the effects of MARK2 phosphorylation of CAMSAP2 on Golgi architecture through expression of WT-CAMSAP2 and two CAMSAP2 S835 mutants in CAMSAP2 KO cells. They find that CAMSAP2 S835A (non-phosphorylatable) was less capable of rescuing Golgi morphology than CAMSAP2 S835D (phosphomimetic). Golgi area has been measured to demonstrate this phenomenon. Representative immunofluorescence images in Fig. 4D appear to indicate that this is the case. However, quantification in Fig. 4E does not show significance between HA-CAMSAP2 and HA-CAMSAP2A that would support the initial claim. The authors could analyze other aspects of Golgi morphology (e.g. number of Golgi fragments, degree of dispersal around the nucleus) to capture the clear structural defects demonstrated in HACAMSAP2A cells.

We sincerely thank the reviewer for their valuable feedback and for pointing out potential areas of improvement in our analysis of Golgi morphology. We apologize for any misunderstanding caused by our description of the results in Figure 4E.

The quantification indeed shows a significant difference between HA-CAMSAP2 and HACAMSAP2A in terms of Golgi area, as indicated in the figure by the statistical annotations (pvalue provided in the legend). To ensure clarity, we have revised the figure legend (page 32, lines 19-23 in the revised manuscript) to explicitly describe the statistical significance, and the method used for quantification.

Because the quantification indeed shows a significant difference between HA-CAMSAP2 and HA-CAMSAP2A in terms of Golgi area, and to maintain consistency throughout the manuscript, we did not further analyze other aspects of Golgi morphology.

We hope this clarification, along with the additional analyses, will address the reviewer’s concerns. Once again, we are deeply grateful for these constructive comments, which have helped us improve the quality and robustness of our study.

- Wound edge assays are used to capture the difference in Golgi reorientation towards the leading edge between CAMSAP2 S835A and CAMSAP2 S835D. However, these studies lack comparison to WT-CAMSAP2 that would support the role of phosphorylated CAMSAP2 in reorienting the Golgi in this context.

We sincerely thank the reviewer for their insightful suggestion. In response, we have added a comparison between CAMSAP2 S835A/D and WT-CAMSAP2, in addition to HT1080 and MARK2 KO cells, to better evaluate the role of phosphorylated CAMSAP2 in Golgi reorientation.

The results, now shown in Figure 5A-C, indicate that in the absence of MARK2, there is no significant difference in Golgi reorientation between WT-CAMSAP2 and CAMSAP2 S835A. This observation supports the conclusion that MARK2-mediated phosphorylation of CAMSAP2 at S835 is essential for effective Golgi reorientation.

To enhance clarity, we have updated the corresponding Results section (page 9, lines 37-40 and page 10, line 1 in the revised manuscript) to describe this additional comparison. We believe this analysis strengthens our findings and provides a clearer understanding of the role of phosphorylated CAMSAP2 in Golgi dynamics.

We hope this additional data addresses the reviewer’s concerns. Once again, we are grateful for the constructive feedback, which has helped improve the clarity and robustness of our study.

(4) Identification of CAMSAP2 interaction partners- Quantification of interaction ability between CAMSAP2 and CG-NAP, CLASP2, or USO1 in Fig. 5D, 5F and 5J respectively, lack WT-CAMSAP2 comparisons.

We sincerely thank the reviewer for their valuable suggestion. In response, we have included WT-CAMSAP2 data in the quantification of interaction ability between CAMSAP2 and CG-NAP, CLASP2, and USO1. These results, now shown in revised Figures 5 D-G and Figures 6 C-D, provide a direct comparison that further validates the differential interaction abilities of CAMSAP2 mutants.

The inclusion of WT-CAMSAP2 allows us to better contextualize the effects of specific mutations on CAMSAP2 interactions and strengthens our conclusions regarding the role of these interactions in Golgi dynamics.

We hope this addition addresses the reviewer’s concerns and enhances the clarity and robustness of our study. We deeply appreciate the constructive feedback, which has been instrumental in improving our manuscript.

- The CG-NAP immunoblot presented in Fig. 5C shows that the protein is 310 kDa, which is the incorrect molecular weight. CG-NAP (AKAP450) should appear at around 450 kDa. Further, no CG-NAP antibody is included in Table 2 - Information of Antibodies. The authors need to explain this discrepancy.

We sincerely apologize for the lack of clarity in our annotation and description, which may have caused confusion regarding the CG-NAP immunoblot presented in Figure 5C (Figure 5D in the revised manuscript). To clarify, CG-NAP (AKAP450) is indeed a 450 kDa protein, and the marker at 310 kDa represents the molecular weight marker’s upper limit, above which CG-NAP is observed. This has been clarified in the figure legend (page 33, lines 21-23 in the revised manuscript).

Regarding the CG-NAP antibody, it was custom-made and purified in our laboratory. Polyclonal antisera against CG-NAP, designated as αEE, were generated by immunizing rabbits with GSTfused fragments of CG-NAP (aa 423–542). This antibody has been validated extensively in our previous research, demonstrating its specificity and reliability (Wang et al., 2017). The details of the antibody preparation are included in the footnote of Table 2 for reference.

We hope this clarification, along with the additional context regarding the antibody validation, resolves the reviewer’s concerns. We are deeply grateful for the reviewer’s attention to detail, which has helped us improve the clarity and rigor of our manuscript.

References：

Wang, J., Xu, H., Jiang, Y., Takahashi, M., Takeichi, M., and Meng, W. (2017). CAMSAP3dependent microtubule dynamics regulates Golgi assembly in epithelial cells. Journal of genetics and genomics = Yi chuan xue bao 44 (1): 39-49.

Minor Comments- Authors should change immunofluorescence images to colorblind friendly colors. The current presentation of merged overlays makes it really difficult to interpret- I would strongly encourage inverted or at a minimum greyscale individual images of key proteins of interest.

We sincerely thank the reviewer for their valuable suggestion regarding the presentation of immunofluorescence images. In response, we have converted the images in Figure 1C to greyscale individual images for each key protein of interest. This adjustment ensures that the figures are more accessible and interpretable, including for readers with color vision deficiencies.

We hope this modification addresses the reviewer’s concern and improves the clarity of our data presentation. We are grateful for the constructive feedback, which has helped us enhance the overall quality of our figures.

- On p. 8 text should be amended to 'Previous literature has documented MARK2's localization to the microtubules, microtubule-organizing center (MTOC), focal adhesions..'

We sincerely thank the reviewer for their comment regarding the text on page 8. Considering the reasoning provided in response to question 2, where we clarified that MARK2's Golgi localization is not fully understood, we have decided to remove this section from the manuscript to maintain the accuracy and rigor of our study.

We appreciate the reviewer’s attention to detail and constructive feedback, which has helped us improve the clarity and focus of our manuscript.

- In Fig.1A scale bars are not shown on individual channel images of CAMSAP or GM130

We sincerely thank the reviewer for pointing out the omission of scale bars in the individual channel images of CAMSAP and GM130 in Figure 1A (Figure 1C in the revised manuscript). In response, we have added a scale bar (5 μm) to the CAMSAP2 channel, as shown in the revised Figure 1C. These updates have been described in the figure legend (page 29, line 21).

We hope this modification addresses the reviewer’s concern and improves the accuracy and clarity of our figure presentation. We greatly appreciate the reviewer’s constructive feedback, which has helped enhance the quality of our manuscript.

- In Fig. 1B the title should be amended to 'Colocalization of CAMSAP2/GM130'

We sincerely thank the reviewer for their suggestion to amend the title in Figure 1B (Figure 1D in the revised manuscript). In response, we have updated the title to "Colocalization of CAMSAP2/GM130," as shown in the revised Figure 1D.

We hope this modification addresses the reviewer’s concern and improves the clarity and accuracy of the figure. We greatly appreciate the reviewer’s valuable feedback, which has helped us refine the presentation of our results.

- In Fig. 2F, 5A, and Sup Fig 3C scale bars have been presented vertically

We sincerely thank the reviewer for pointing out the issue with the vertical orientation of scale bars in Figures 2F (Figure 2D in the revised manuscript), 5A, and Supplementary Figure 3C. In response, we have modified the scale bars in revised Figures 2D and 5A to a horizontal orientation for improved consistency and clarity. Additionally, Supplementary Figure 3C has been removed from the revised manuscript.

We hope these adjustments address the reviewer’s concerns and enhance the overall presentation quality of the figures. We greatly appreciate the reviewer’s constructive feedback, which has helped us refine our manuscript.

- Panels are not correctly aligned, and images are not evenly spaced or sized in multiple figures - Fig. 2F, 4D, Sup Fig. 1F, Sup Fig. 2C, Sup Fig. 3E, Sup Fig. 4C

We sincerely thank the reviewer for pointing out the misalignment and uneven spacing or sizing of panels in multiple figures, including Figures 2F, 4D, Supplementary Figures 1F, 2C, 3E, and 4C (Figure 2D, 4D, Supplementary Figures 1F, 2C, and 3H in the revised manuscript. Supplementary Figure 3E was removed from our manuscript). In response, we have standardized the spacing and sizing of all panels throughout the manuscript to ensure consistency and improve visual clarity.

We hope this modification addresses the reviewer’s concerns and enhances the overall presentation quality of our figures. We greatly appreciate the reviewer’s constructive feedback, which has helped us improve the organization and professionalism of our manuscript.

- An uncolored additional data point is present in Fig. 3F

We sincerely thank the reviewer for pointing out the presence of an uncolored additional data point in Figure 3F. In response, we have removed this data point from the revised figure to ensure accuracy and clarity.

We hope this adjustment resolves the reviewer’s concern and improves the overall quality of the figure. We greatly appreciate the reviewer’s careful review and constructive feedback, which have helped us refine our manuscript.

- In Fig. 3A 'GAMSAP2/GM130' in the vertical axis label should be amended to 'CAMSAP2/GM130'

We sincerely thank the reviewer for pointing out the error in the vertical axis label of Figure 3A. In response, we have corrected "GAMSAP2/GM130" to "CAMSAP2/GM130," as shown in the revised Figure 3I.

We hope this correction resolves the reviewer’s concern and improves the accuracy of our figure. We greatly appreciate the reviewer’s careful review and constructive feedback, which have helped us refine our manuscript.

- In Fig 5A the green label should be amended to 'GFP-CAMSAP2' instead of 'GFP'

We sincerely apologize for the confusion caused by our labeling in Figure 5A. To clarify, the green label “GFP” refers to the antibody used, while “GFP-CAMSAP2” is indicated at the top of the figure to specify the construct being analyzed.

We hope this explanation resolves the misunderstanding and provides clarity regarding the labeling in Figure 5A. We greatly appreciate the reviewer’s feedback, which has allowed us to address this issue and improve the precision of our figure annotations.

- The repeated use of contractions throughout the manuscript was distracting, I would strongly encourage removing these.

We sincerely thank the reviewer for pointing out the distracting use of contractions in the manuscript. In response, we have removed and replaced all contractions with their full forms to improve the clarity and formal tone of the text.

We hope this modification addresses the reviewer’s concern and enhances the readability and professionalism of our manuscript. We greatly appreciate the reviewer’s constructive feedback, which has helped us refine the quality of our writing.

**Reviewer #2:**
SummaryThis work by the Meng lab investigates the role of the proteins MARK2 and CAMSAP2 in the Golgi reorientation during cell polarisation and migration. They identified that both proteins interact together and that MARK2 phosphorylates CAMSAP2 on the residue S835. They show that the phosphorylation affects the localisation of CAMSAP2 at the Golgi apparatus and in turn influences the Golgi structure itself. Using the TurboID experimental approach, the author identified the USO1 protein as a protein that binds differentially to CAMSAP2 when it is itself phosphorylated at residue 835. Dissecting the molecular mechanisms controlling Golgi polarisation during cell migration is a highly complex but fundamental issue in cell biology and the author may have identified one important key step in this process. However, although the authors have made a genuine iconographic effort to help the reader understand their point of view, the data presented in this study appear sometimes fragile, lacking rigour in the analysis or over-interpreted. Additional analyses need to be conducted to strengthen this study and elevate it to the level it deserves.

We sincerely thank the reviewer for their thoughtful evaluation and recognition of our study's significance in understanding Golgi reorientation during cell migration. We appreciate the constructive feedback regarding data robustness, clarity, and interpretation. In response, we have conducted additional analyses, revised data presentation, and ensured cautious interpretation throughout the manuscript. These changes aim to address the reviewer’s concerns comprehensively and strengthen the scientific rigor of our study.

Major commentsIn order to conclude as they do about the putative role of USO1, the authors need to perform a siRNA/CRISPR of USO1 to validate its role in anchoring CAMSAP2 to the Golgi apparatus in a MARK2 phosphorylation-dependent manner. In other words, does depletion of USO1 affect the recruitment of CAMSAP2 to the Golgi apparatus?

We sincerely thank the reviewer for their insightful suggestion regarding the role of USO1 in anchoring CAMSAP2 to the Golgi apparatus. In response, we performed USO1 knockdown using siRNA and quantified the Pearson correlation coefficient of CAMSAP2 and GM130 colocalization in control and USO1-knockdown cells.

The results show that CAMSAP2 localization to the Golgi is significantly reduced in USO1knockdown cells, confirming that USO1 plays a critical role in recruiting CAMSAP2 to the Golgi apparatus. These results are now presented in Figures 6 E–G, and corresponding updates have been incorporated into the Results section (page 10, lines 36-37 in the revised manuscript).

We hope this additional experiment addresses the reviewer’s concern and strengthens our conclusions regarding the role of USO1. We are grateful for the reviewer’s constructive feedback, which has greatly improved the robustness of our study.

It is not clear from this study exactly when and where MARK2 phosphorylates CAMSAP2. What is the result of overexpression of the two proteins in their respective localisation to the Golgi apparatus? As binding between CAMSAP2 and MARK2 appears robust in the immunoprecipitation assay, this should be readily investigated.

We sincerely thank the reviewer for their insightful comments and questions. To address the role of MARK2 in regulating CAMSAP2 localization to the Golgi apparatus, we overexpressed GFPMARK2 in cells and compared its effects on CAMSAP2 localization to the Golgi with control cells overexpressing GFP alone. Our results show that CAMSAP2 localization to the Golgi is significantly increased in GFP-MARK2-overexpressing cells, as shown in Supplementary Figures 3C and 3E. Corresponding updates have been incorporated into the Results section (page 8, lines 25-27 in the revised manuscript).

Regarding the question of how MARK2 itself localizes to the Golgi, we are currently unable to fully elucidate the underlying mechanism. Therefore, we have removed the discussion of MARK2’s Golgi localization from the manuscript to ensure scientific accuracy. Consequently, we have not conducted experiments to assess the effects of CAMSAP2 overexpression on MARK2’s localization to the Golgi.

We hope this explanation clarifies the reviewer’s concerns. We are grateful for the reviewer’s constructive feedback, which has guided us in improving the clarity and focus of our study.

To strengthen their results, can the author map the interaction domains between CAMSAP2 and MARK2? The authors have at their disposal all the constructs necessary for this dissection.

We sincerely thank the reviewer for their insightful suggestion to map the interaction domains between CAMSAP2 and MARK2. In response, we performed immunoprecipitation experiments using truncated constructs of CAMSAP2. Our results reveal that MARK2 interacts specifically with the C-terminus (1149F) of CAMSAP2, as shown in Supplementary Figures 3A and 3B. Corresponding updates have been incorporated into the Results section (page 7, lines 41-42 and page 8, line 1 in the revised manuscript).

We hope this additional analysis addresses the reviewer’s suggestion and further strengthens our conclusions. We greatly appreciate the reviewer’s constructive feedback, which has helped improve the depth of our study.

Minor commentsSup-fig1H: It is not clear if the polarisation experiment has been repeated three times (as it should) and pooled or is just the result of one experiment?

We sincerely apologize for the lack of clarity regarding the experimental details for Supplementary Figure 1H. To clarify, the polarization experiment was repeated three times, and the results were pooled to generate the data presented. We have updated the figure legend for Supplementary Figure 1H to explicitly state this information (page 35, lines 27-29 in the revised manuscript).

We hope this clarification resolves the reviewer’s concern. We greatly appreciate the reviewer’s careful review and constructive feedback, which have helped us improve the accuracy and transparency of our manuscript.

Sup-fig2C: "Immunofluorescence staining plots" formula used in the legend is not clear. Which condition is presented in the panel, parental HT1080 or CAMSAP2 KO cells?

We thank the reviewer for pointing out the lack of clarity regarding the conditions presented in Supplementary Figure 2C. To clarify, the immunofluorescence staining plots shown in this panel are from parental HT1080 cells. We have updated the figure legend to include this information (page 36, line 14 in the revised manuscript).

We hope this clarification resolves the reviewer’s concern and improves the transparency of our data presentation. We greatly appreciate the reviewer’s feedback, which has helped us refine the manuscript.

Figure 1D: In the plot, the colour of the points for the "red cells" are red but the one for the "blue cells" are green, this is confusing.E: Once again, the colour choice is confusing as blue cells (t=0.5h) are quantified using red dots and red cells (t=2h) quantified using green dots. The t=0h condition should be quantified as well and added to the graph.F: Representative CAMSAP2 immunofluorescence pictures for the three time points should be provided in addition to the drawings.

We thank the reviewer for their valuable comments regarding Figure 1D (revised Figure 1E), Figure 1E (revised Figure 1B), and Figure 1F (revised Supplementary Figure 2C).

- Figure 1D (revised Figure 1E): we have modified the x-axis labels and adjusted the color scheme of the data points to ensure consistency and avoid confusion.

- Figure 1E (revised Figure 1B): we have updated the x-axis and included the quantification of the t=0h condition, which has been added to the graph.

- Figure 1F (revised Supplementary Figure 2C): we have provided representative immunofluorescence images of CAMSAP2 for the three-time points to complement the schematic drawings.

We hope these revisions address the reviewer’s concerns and improve the clarity and completeness of our data presentation. We greatly appreciate the reviewer’s constructive feedback, which has significantly contributed to enhancing our manuscript.

Figure 2A: No methodology in the material and methods is provided for this analysis.B: Can the authors be more precise regarding the source of the CAMSAP2 interactants? Can the author provide the citation of the publication describing the CAMSAP2-MARK2 interaction?D: Genotyping for the MARK2 KO cell line should be provided the same way it was provided for the CAMSAP2 cell line in Sup-fig1. "MARK2 was enriched around the Golgi apparatus in a significant proportion of HT1080 cells": which proportion of the cells?F: The time point of fixation is missingG: It is not clear if the polarisation experiment has been repeated three times (as it should) and pooled or is just the result of one experiment?

We thank the reviewer for their detailed comments and suggestions regarding Figure 2. Below, we provide clarifications and outline the modifications made:

- Figure 2A: The methodology for this analysis has been added to section 5.14 (Data statistics). Specifically, we have stated: “GO analysis of proteins was plotted using https://www.bioinformatics.com.cn, an online platform for data analysis and visualization” (page 26 lines 5-6 in the revised manuscript).

- Figure 2B: The CAMSAP2 interactants were derived from the study by Wu et al., 2016, which provides the source of these interactants. The interaction between CAMSAP2 and MARK2 is referenced from Zhou et al., 2020. These citations have been added to the relevant sections of the manuscript (page 30, lines 10-11 and 13-14).

- Figure 2D (removed in the revised manuscript): Genotyping for the MARK2 KO cell line has been provided in the same format as for the CAMSAP2 KO cell line in Figure 2G. Additionally, as the MARK2 Golgi localization discussion cannot yet be fully elucidated, we have removed this portion from the manuscript.

- Figure 2F (revised Figure 2D): The time point of fixation, which occurred 2 hours after the scratch wound assay, has been added to the figure legend (page 30, lines 15-16).

- Figure 2G (revised Figure 2E-F): The polarization experiment was repeated three times, and the results were pooled. This information has been included in the figure legend (page 30, lines 26 and 29).

We hope these updates address the reviewer’s concerns and improve the clarity and completeness of the manuscript. We are grateful for the reviewer’s constructive feedback, which has greatly enhanced the rigor of our study. References：

Wu, J., de Heus, C., Liu, Q., Bouchet, B.P., Noordstra, I., Jiang, K., Hua, S., Martin, M., Yang, C., Grigoriev, I., et al. (2016). Molecular Pathway of Microtubule Organization at the Golgi Apparatus. Dev Cell 39 (1): 44-60.

Sup-fig3E: Although colocalisation between CAMSAP2 and MARK2 is clear in your serum conditions in HT1080 and RPE1 cells, the deletion domain analysis appears weak and insufficient to implicate the role of the spacer domain. This part should be deleted or strengthened, but the data do not satisfactorily support your conclusion as it stands.

We sincerely thank the reviewer for their critical comments regarding the deletion domain analysis of MARK2 and its role in colocalization with CAMSAP2. As the current data do not satisfactorily support our conclusions, we have removed all related content on MARK2 and the deletion domain analysis from the manuscript to maintain scientific rigor.

We appreciate the reviewer’s valuable feedback, which has helped us refine and improve the quality and focus of our study.

Figure 3A: Can the reduced CAMSAP2 Golgi localisation phenotype be rescued by the overexpression of MARK2 cDNA in the MARK2 KO cells?F: Presence of a white dot on the HT1080 plotG: The composition of the homogenization buffer is not indicated in the material and methods

We thank the reviewer for their valuable comments and suggestions regarding Figure 3. Below, we detail the modifications made:

- Figure 3A: To address whether the reduced CAMSAP2 Golgi localization phenotype can be rescued, we overexpressed MARK2 cDNA in MARK2 KO cells. Our results show that overexpression of MARK2 successfully rescues the reduced CAMSAP2 localization to the Golgi, as demonstrated in Supplementary Figures 3C and 3E (page 8, lines 5-7).

- Figure 3F: We have removed the white dot on the HT1080 plot to ensure clarity and accuracy.

- Figure 3G: The composition of the homogenization buffer used in the experiment has been added to the Materials and Methods section for completeness (page 24, lines 34-41 and page 25, lines 1-10).

We hope these revisions address the reviewer’s concerns and enhance the clarity and rigor of our study. We are grateful for the reviewer’s constructive feedback, which has significantly improved the quality of our manuscript.

Figure 4B: Quantification of the effect of the S835A mutation should be providedD: Top left panel: Why Ha antibody stains Golgi structure in absence of Ha-CAMSAP2 transfection ? IF the Ha antibody has unspecific affinity towards the Golgi apparatus, may be it is not the good tag to use in this assay?E: The number of cells studied should be standardized. 119 cells were analyzed in the CAMSAP KO vs only 35 cells in the CAMSAP2 KO (HA-CAMSAP2-S835D) conditions. This could introduce strong bias to the analysis. Furthermore the CAMSAP2 S835A seems to provide a certain level of rescue. It would be interesting to see what is the result of the T test between the HT1080 and HA-CAMSAP S835A conditions.

We thank the reviewer for their thoughtful comments and suggestions regarding Figure 4. Below, we detail the revisions and clarifications made:

- Figure 4B: The S835A mutation renders CAMSAP2 non-phosphorylatable by MARK2. This conclusion is based on our experimental observations and previously reported mechanisms.

- Figure 4D: The HA antibody does not exhibit non-specific affinity toward the Golgi apparatus. The observed labeling in the top left panel was due to an error in our annotation. We have corrected the label, replacing "HA" with "CAMSAP2" to accurately reflect the experimental conditions.

- Figure 4E: To standardize the number of cells analyzed across conditions, we reduced the number of CAMSAP2 KO cells analyzed to 50 and balanced the sample sizes for comparison. Additionally, we performed a t-test between the HT1080 and HACAMSAP2 S835A conditions. The results support that CAMSAP2 S835A provides partial rescue, as reflected in the updated analysis (page 32, lines 19-23).

We hope these revisions address the reviewer’s concerns and improve the accuracy and reliability of our results. We greatly appreciate the reviewer’s constructive feedback, which has significantly enhanced the quality of our study.

Figure 66A: The wound position should be indicated on the picture.6B: Given that microtubule labelling is present on the vast majority of the cell surface, this type of quantification provides very little information using conventional light microscopy and should not be used to conclude any change in the microtubule network using Pearson's coefficient. The text describing the figure 6A and 6B needs re written as I do not understand what the author want to say. "In cells located before the wound edge..." : I do not understand how a cell could be located before the wound edge. Which figure corresponds to the trailing edge of the wounding?

We thank the reviewer for their valuable comments on Figure 6A (revised Supplementary Figure 6E) and Figure 6B (revised Supplementary Figure 6F). Below, we detail the modifications made:

- Figure 6A (revised Supplementary Figure 6E), we have added arrows to indicate the wound position, providing clearer guidance for interpreting the image.

- Figure 6B (revised Supplementary Figure 6F), we revised our quantification method based on the approach used in literature (Wu et al., 2016). Specifically, we analyzed the relationship between microtubules and the Golgi apparatus in cells at the leading edge of the wound. The x-axis represents the distance from the Golgi center, while the y-axis shows the normalized radial fluorescence intensity of microtubules and the Golgi apparatus.

Additionally, we revised the accompanying text for clarity and accuracy. The original description:

“In cells located before the wound edge, the Golgi apparatus maintained a ribbon-like shape, with a higher density of microtubules. In contrast, at the trailing edge of the wounding, the Golgi apparatus appeared more as stacks around the nucleus, with fewer microtubules” was replaced with:

“Finally, to comprehensively understand the dynamics between non-centrosomal microtubules and the Golgi apparatus during Golgi reorientation, we conducted cell wound-healing experiments (Supplementary Figure 6 E-F). Our observations revealed notable changes in the Golgi apparatus and microtubule network distribution in relation to the wounding. These findings corroborate our earlier results and suggest a highly dynamic interaction between the Golgi apparatus and microtubules during Golgi reorientation” (Revised manuscript page 11 lines 3-10).

We hope these changes address the reviewer’s concerns and improve the clarity and robustness of our study. We greatly appreciate the reviewer’s constructive feedback, which has significantly enhanced the presentation and interpretation of our data. References：

Wu, J., de Heus, C., Liu, Q., Bouchet, B.P., Noordstra, I., Jiang, K., Hua, S., Martin, M., Yang, C., Grigoriev, I., et al. (2016). Molecular Pathway of Microtubule Organization at the Golgi Apparatus. Dev Cell 39 (1): 44-60.

**Reviewer #3:**
SummaryIn this study, Xu et al. analyzed the wound healing process of HT1080 cells to elucidate the molecular mechanisms by which the Golgi apparatus exhibits transient dispersion before reorienting to the wound edge in the compact assembly structure. They focused on the role of the microtubule minus-end binding protein CAMSAP2, which mediates the linkage between microtubules and the Golgi membrane. At first, they noticed that CAMSAP2 transiently lost Golgi colocalization during the initial phase of the wound healing process. They further found that the cell polarity-regulating kinase MARK2 binds and phosphorylates S835 of CAMSAP2, thereby enhancing the interaction between CAMSAP2 and the Golgi protein Uso1. Together with the phenotypes of CAMSAP2, MARK2, and Uso1 KO cells, these authors argue that the MARK2dependent phosphorylation of CAMSAP2 plays an important role in the reassembly and reorientation of the Golgi apparatus after a transient dispersion observed during the wound healing process.

We sincerely thank the reviewer for their thoughtful summary of our study and constructive feedback. Your comments have been invaluable in refining our research and enhancing the clarity and impact of our manuscript.

Major comments(1) The premise of this study was that during the wound healing process, the Golgi apparatus exhibits transient dispersion before reorientation to the front of the nucleus.In the first place, this claim has not been well established in previous studies or this paper. Therefore, the authors should present a proof of this claim in a clearer manner.To introduce this cellular event, the authors cite several papers in the introduction (page 4) and the results (page 6) sections. However, many papers cited are review articles, and some of them do not describe this change in the Golgi assembly structure before reorientation. Only two original articles discussed this phenomenon (Bisel et al. 2008 and Wu et al. 2016), and direct evidence was provided by only one paper (Wu et al. 2016) in which changes in the Golgi apparatus in wound-healing RPE1 cells were recorded by live imaging (Fig.7A in Wu et al. 2016).Furthermore, it should be noted that this previous paper demonstrated that depletion of CAMSAP2 inhibits Golgi dispersion. Obviously, this conclusion is inconsistent with their statement to introduce this study (page4) that ‟This emphasizes CAMSAP2's role in sustaining Golgi integrity during critical cellular events like migration." In addition, it also contradicts the authors' model of the present paper (Fig. 6E), which argued that disruption of the Golgi association of CAMSAP2 facilitates the Golgi dispersion.

We sincerely thank the reviewer for their detailed comments and for providing us with the opportunity to clarify the premise and conclusions of our study. Below, we address the main concerns raised:

First, to provide direct evidence of Golgi apparatus changes during the wound-healing process, we conducted live-cell imaging experiments. Our observations, presented in revised Supplementary Figure 2A, clearly demonstrate that the Golgi apparatus exhibits a transient dispersion state before reorienting toward the leading edge of the nucleus during migration.

Regarding the interpretation of previous studies, we acknowledge the reviewer’s concerns about the citation of review articles. To address this, we have revisited the literature and clarified that the phenomenon of Golgi dispersion during reorientation has been directly demonstrated in Wu et al (Wu et al., 2016), where live imaging of wound-healing RPE1 cells showed this dynamic behavior. Furthermore, we note that in Wu et al paper explicitly demonstrates that CAMSAP2 depletion promotes Golgi dispersion, contrary to the reviewer’s interpretation that "depletion of CAMSAP2 inhibits Golgi dispersion."

Our model focuses on the role of CAMSAP2 in restoring the Golgi from a transiently dispersed structure back to an intact ribbon-like structure during reorientation. Specifically, we propose that during this process, the disruption of CAMSAP2’s association with the Golgi affects this restoration, rather than directly promoting Golgi dispersion as suggested by the reviewer. We believe this distinction aligns with our data and the existing literature.

To strengthen the background of our study, we have revised the introduction and results sections (page 6, lines 6-13 and page 7, lines 1-17) to minimize reliance on review articles and have provided more explicit citations to original research papers. We hope this addresses the reviewer’s concern about the sufficiency of the cited literature.

We trust these clarifications and revisions resolve the reviewer’s concerns and enhance the robustness of our study. Once again, we are grateful for the reviewer’s constructive feedback, which has greatly helped refine our manuscript. References：

Wu, J., de Heus, C., Liu, Q., Bouchet, B.P., Noordstra, I., Jiang, K., Hua, S., Martin, M., Yang, C., Grigoriev, I., et al. (2016). Molecular Pathway of Microtubule Organization at the Golgi Apparatus. Dev Cell 39 (1): 44-60.

The authors did not provide experimental data for this temporal change in the Golgi assembly structures during the wound-healing process of HT1080 that they analyzed. They only provide an illustration of wound-healing cells (Fig.1F), in which cells are qualitatively discriminated and colored based on the Golgi states, without indicating the experimental basis of the discrimination.According to their ambiguous descriptions in the text (page7), the reader can speculate that Fig. 1F is illustrated based on the images in Supplementary Fig. 2C. However, because of the low quality and presentation style of these data, it is impossible to recognize the assembly structures of the Golgi apparatus in wound-edge cells.If the authors hope to establish this premise claim for the present paper, they should provide their own data corresponding to the present Supplementary Fig. 2C in more clarity and present qualitative data verifying this claim, as Wu et al. did in Fig. 7A in their paper.

We sincerely thank the reviewer for their constructive feedback and the opportunity to address the concern regarding the lack of experimental data supporting the temporal changes in Golgi assembly during the wound-healing process.

To establish this premise, we conducted live-cell imaging experiments to observe the dynamic changes in the Golgi apparatus during directed cell migration. Our data, now presented in Supplementary Figure 2A, clearly demonstrate that the Golgi apparatus undergoes a transient dispersed state before reorganizing into an intact structure. These findings provide direct experimental evidence supporting our claim.

In addition, we have revised the data originally presented in Supplementary Figure 2C and enhanced its quality and presentation style. This supplementary figure now includes clearer images and annotations to better illustrate the Golgi assembly structures in wound-edge cells. The improved data presentation aligns with the standards set by Wu et al reported (Wu et al., 2016) and provides qualitative support for our observations.

We hope these additions and revisions address the reviewer’s concerns and strengthen the scientific rigor and clarity of our manuscript. We are grateful for the reviewer’s valuable suggestions, which have significantly improved the quality of our study. References：

Wu, J., de Heus, C., Liu, Q., Bouchet, B.P., Noordstra, I., Jiang, K., Hua, S., Martin, M., Yang, C., Grigoriev, I., et al. (2016). Molecular Pathway of Microtubule Organization at the Golgi Apparatus. Dev Cell 39 (1): 44-60.

(2) In Fig.1A-D, the authors claim that CAMSAP2 dissociates from the Golgi apparatus in cells "that have not yet completed Golgi reorientation and exhibit a transitional Golgi structure, characterized by relative dispersion and loss of polarity (page7)." However, I these analyses, they do not analyze the initial stage (0.5h after wound addition) of cells facing the wound edge, as they do in Supplementary Fig. 2C. Instead, they analyze cells separated from the wound edge at 2 h after wound addition when the wound-edge cells complete their polarization. These data are highly misleading because there is no evidence that the cells separated from the wound edge are really in the transitional state before polarization.In this regard, Fig. 1E shows the analysis of the wound-edge cells at 0.5 and 2 h after the addition of wound, which provides suitable data to verify the authors' claim. However, the corresponding legend indicates that these statistical data are based on the illustration in Fig. 1F, which is probably based on highly ambiguous data in Supplementary Fig. 2C (see above).Taken together, I strongly recommend the authors to remove Fig.1A-D. Instead, they should include the improved figure corresponding to the present Supplementary Fig.2C and present its statistical analysis similar to the present Fig.1E for this claim.

We sincerely thank the reviewer for their constructive feedback and recommendations. Below, we address the concerns raised regarding Figure 1A-D and Supplementary Figure 2C.

To provide stronger evidence for the transitional state of the Golgi apparatus during reorientation and the dynamic regulation of CAMSAP2 localization, we conducted live-cell imaging experiments. These results, now presented in Supplementary Figure 2A, clearly demonstrate that the Golgi apparatus undergoes a transitional state characterized by dispersion before reorienting toward the leading edge.

Additionally, we analyzed fixed wound-edge cells at different time points during directed migration to observe CAMSAP2’s colocalization with the Golgi apparatus. The results, shown in Figures 1A and 1B, reveal dynamic changes in CAMSAP2 localization, confirm its regulation during Golgi reorientation, and include a corresponding statistical analysis (page 7, lines 1-17).

These updates ensure that our claims are supported by robust and unambiguous data.

We hope these revisions address the reviewer’s concerns and provide clear and reliable evidence for the transitional state of the Golgi apparatus and CAMSAP2’s dynamic regulation. We are grateful for the reviewer’s constructive suggestions, which have greatly improved the quality and focus of our manuscript.

(3) In Supplementary Fig. 5 and Fig. 4, the authors claim that MARK2 phosphorylates S835 of CAMSAP2.There are many issues to be addressed. Otherwise, the above claim cannot be assumed to be reliable.First, the descriptions (in the text and method sections) and figures (Supplementary Fig.5) concerning the in vitro kinase assay and subsequent phosphoproteomic analysis are too immature and contain many errors.Legend to Supplementary Fig. 5 is too immature for comprehension. It should be completely rewritten in a more comprehensive manner. The figure in Supplementary Fig. 5C is also too immature for understanding. They simply paste raw mass spectrometric data without any modification for presentation.

We sincerely apologize for the lack of clarity and inaccuracies in the original descriptions and figure legends for the in vitro kinase assay and phosphoproteomic analysis. We greatly appreciate the reviewer’s detailed comments, which have allowed us to address these issues comprehensively.

To improve clarity and accuracy, we have rewritten the figure legend for the original Supplementary Figure 5 (now Supplementary Figure 4) as follows:

(A): CBB staining of a gel with GFP-CAMSAP2, GST, and GST-MARK2. GFP-CAMSAP2 was expressed in Sf9 cells and purified. GST and GST-MARK2 were expressed in *E. coli* and purified.

(B): Western blot analysis of an in vitro kinase assay. GST or GST-MARK2 was incubated with GFP-CAMSAP2 in kinase buffer (50 mM Tris-HCl pH 7.5, 12.5 mM MgCl2, 1 mM DTT, 400 μM ATP) at 30°C for 30 minutes. Reactions were stopped by boiling in the loading buffer.

(C): Detection of phosphorylation at S835 in CAMSAP2 by mass spectrometry. The observed mass increases in b4, b5, b6, b7, b8, b10, b11, and b12 fragments indicate phosphorylation at Ser835.

(D): Kinase assay samples analyzed using Phos-tag SDS-PAGE. HEK293 cells were cotransfected with the indicated plasmids. Band shifts of CAMSAP2 mutants were examined via western blot. Phos-tag was used in SDS-PAGE, and arrowheads indicate the shifted bands caused by phosphorylation.

To address the reviewer’s concern about Supplementary Figure 5C, we have reformatted the mass spectrometry data to improve readability and presentation quality. The revised figure includes clearer annotations and graphical representations of the mass spectrometric evidence for phosphorylation at S835.

We believe these updates enhance the comprehensibility and reliability of our data, providing robust support for our claim that MARK2 phosphorylates CAMSAP2 at S835. We hope these

revisions address the reviewer’s concerns and demonstrate our commitment to improving the quality of our manuscript.

The readers cannot understand how the authors purified GFP-CAMSAP2 for the kinase assay.The method section incorrectly states that the product was purified using Ni-resin.

We thank the reviewer for their comment regarding the purification of GFP-CAMSAP2 for the kinase assay. We would like to clarify that GFP-CAMSAP2 carries a His-tag, which allows for purification using Ni-resin, as described in the Methods section (page 23, Lines 32-40). Therefore, the description in the Methods section is correct.

To avoid any potential misunderstanding, we have revised the Methods section to provide more detailed and precise descriptions of the purification process. Specifically, GFP-CAMSAP2 was cloned into the pOCC6_pOEM1-N-HIS6-EGFP vector, which includes a His-tag, and was expressed in Sf9 cells. The His-GFP-CAMSAP2 protein was purified using Ni-resin chromatography. Relevant details have been added to the Methods section (page 21, Lines 34-36: “CAMSAP2 was cloned into the pOCC6_pOEM1-N-HIS6-EGFP vector expressed in Sf9, purified as His-GFP-CAMSAP2.”; page 23, Lines 32-33: “His-GFP-CAMSAP2 was cotransfected with bacmids into Sf9 cells to generate the passage 1 (P1) virus.”).

We hope these clarifications and revisions address the reviewer’s concern and improve the comprehensibility of our experimental details. We appreciate the reviewer’s feedback, which has helped us refine the manuscript.

In this relation, GST and GST-MARK2 are described as having been purified from Sf9 insect cells in the text section (page9) and legend to Supplementary Fig. 5, but from *E. coli* in the method section. Which is correct?

We thank the reviewer for pointing out the inconsistencies in the descriptions regarding the source of GST and GST-MARK2. To clarify, both GST and GST-MARK2 were purified from *E. coli*, as stated in the Methods section (page 23, Lines 26-31). We have corrected the erroneous descriptions in the main text (page 8, Lines 35-36) and the legend to Supplementary Figure 4 to ensure consistency.

Additionally, we have updated the legend for Supplementary Figure 4A to state the sources of each protein explicitly:

“GFP-CAMSAP2 were expressed in Sf9 cells and purified. GST and GST-MARK2 were expressed in *E. coli* and purified.” (page 38, Lines 2-3)

These revisions ensure that the experimental details are accurate and consistent across the manuscript, eliminating any potential confusion. We appreciate the reviewer’s careful review and constructive feedback, which have helped us improve the clarity and reliability of our study.

Because the phosphoproteomic data (Supplementary Fig. 5C) are not provided clearly, the experimental data for Fig.4A, in which possible CAMSAP2 phosphorylation sites are illustrated, are completely unknown. For me, it is highly strange that only the serine residues are listed in Fig. 4A.

We sincerely thank the reviewer for raising this important point regarding Figure 4A and the phosphoproteomic data in Supplementary Figure 5C.

- Phosphorylation Sites in Figure 4A

The phosphorylation sites illustrated in Figure 4A are derived from our analysis of the original mass spectrometry data. These sites were included based on their high confidence scores and data reliability. Importantly, only serine residues met the stringent criteria for inclusion, as no threonine or tyrosine residues had sufficient evidence for phosphorylation. To clarify this, we have updated the figure legend for Figure 4A (page 32, Lines3-7).

- Improvements to Supplementary Figure 5C (Supplementary Figure 4D in the revised manuscript)

To enhance transparency and clarity, we have reformatted Supplementary Figure 4D to include clearer annotations. The revised figure highlights the phosphopeptides used to identify the phosphorylation sites and provides a more comprehensive presentation of the mass spectrometry data. To clarify this, we have updated the figure legend for Supplementary Figure 4D (page 38, Lines 11-13).

- Data Availability

We will follow the journal’s guidelines by uploading the raw mass spectrometry data to the required public database upon manuscript acceptance. This ensures that the data are accessible and reproducible in compliance with journal standards.

We hope these clarifications and updates address the reviewer’s concerns and improve the reliability and comprehensibility of our data presentation. We greatly appreciate the reviewer’s constructive feedback, which has helped us enhance the rigor and clarity of our manuscript.

Considering the crude nature of the GST-MARK2 sample used for the in vitro kinase assay (Supplementary Fig. 5A), it is unclear whether MARK2 is responsible for all phosphorylation sites on CAMSAP2 detected in the phosphoproteomic analysis. Furthermore, if GFP-CAMSAP2 was purified from Sf9 insect cells, these sites might have been phosphorylated before incubation for the in vitro kinase assay. The authors should address these issues by including a negative control using the kinase-dead mutant of MARK2 in their in vitro kinase assay.

We sincerely thank the reviewer for raising these important points regarding the potential prephosphorylation of GFP-CAMSAP2 and the role of MARK2 in the phosphorylation sites detected in our analysis.

To address the possibility that GFP-CAMSAP2 may have been pre-phosphorylated during its expression in Sf9 insect cells, we conducted an in vitro comparison. Specifically, we compared the band shifts observed in GST-MARK2 + GFP-CAMSAP2 versus GST + GFP-CAMSAP2 under identical conditions. As shown in Supplementary Figure 4B, the GST-MARK2 + GFP-CAMSAP2 group exhibited a clear upward band shift compared to the GST + GFP-CAMSAP2 group, indicating additional phosphorylation events induced by MARK2.

Regarding the inclusion of a kinase-dead MARK2 mutant as a negative control, we acknowledge this as a valuable suggestion for further confirming the specificity of MARK2 in phosphorylating CAMSAP2. While this experiment is not currently included, we plan to conduct it in our future studies to strengthen our findings.

We hope this clarification and the provided evidence address the reviewer’s concerns. We are grateful for this constructive feedback, which has helped us critically evaluate and refine our experimental approach.

(4) In Supplementary Fig.6A-C and Fig.5A-B, the authors claim that the phosphorylation of CAMSAP2 S835 is required for restoring the reduced reorientation of the Golgi in wound-healing cells and the delay in wound closure observed in MARK2 KO cells.If the aforementioned claim is adequately supported by experimental data, it indicates that the defects in Golgi repolarization and wound closure in MARK2 KO cells can be mainly attributed to the reduced phosphorylation of S835 of CAMSAP2 in HT1080. Considering the presence of many well-known substrates of MARK2 for regulating cell polarity, this claim is highly striking.However, to strongly support this conclusion, the authors should first perform a rescue experiment using MARK2 KO cells exogenously expressing MARK2. This step is essential for determining whether the defects observed in MARK2 KO cells are caused by the loss of MARK2 expression, but not by other artificial effects that were accidentally raised during the generation of the present MARK2 KO clone.

We sincerely thank the reviewer for their insightful suggestion regarding the rescue experiment to confirm that the defects observed in MARK2 KO cells are specifically caused by the loss of MARK2 expression.

To address this, we performed a rescue experiment in MARK2 KO HT1080 cells by exogenously expressing GFP-MARK2. Our results, presented in Supplementary Figures 3C-E, demonstrate that GFP-MARK2 expression successfully restores the localization of CAMSAP2 on the Golgi apparatus in MARK2 KO cells.

These findings strongly support the conclusion that the defects in Golgi architecture and CAMSAP2 Golgi localization are directly attributable to the loss of MARK2 expression, rather than any artificial effects potentially introduced during the generation of the MARK2 KO clone.

We hope these additional experimental results address the reviewer’s concerns and provide robust evidence for the role of MARK2 in regulating Golgi reorientation and wound closure. We are grateful for the reviewer’s constructive feedback, which has significantly improved the rigor and clarity of our study.

In addition, to evaluate the impact of the rescue effect of CAMSAP2, the authors should include the data of wild-type HT1080 and MARK2 KO cells in Fig. 5B to reliably demonstrate the aforementioned claim.

We thank the reviewer for their valuable suggestion to include data from wild-type HT1080 and MARK2 KO cells in Figure 5A-C to better evaluate the rescue effects of CAMSAP2.

In response, we have incorporated data from wild-type HT1080 and MARK2 KO cells into Figure 5A-C. These additions provide a comprehensive comparison and further demonstrate the impact of CAMSAP2-S835A and CAMSAP2-S835D on Golgi reorientation relative to the wild-type and MARK2 KO conditions.

These changes are reflected in Figures 5A-C.

We hope these updates address the reviewer’s concerns and strengthen the reliability of our conclusions. We greatly appreciate the reviewer’s constructive feedback, which has significantly enhanced the robustness of our study.

Principally, before checking the rescue effects in MARK2 KO cells, the authors should examine the rescue activity of the CAMSAP2 S835 mutants in restoring the reduced reorientation of the Golgi in wound-healing cells and the delay in wound closure observed in CAMSAP2 KO cells (Supplementary Fig.1F-H and Supplementary Fig.2A, B). These experiments are more essential experiments to substantiate the authors' claim.

We thank the reviewer for their insightful suggestion to examine the rescue activity of CAMSAP2 S835 mutants in CAMSAP2 KO cells to further substantiate our claims.

In Figure 4D-F, we observed significant differences between CAMSAP2 S835 mutants in their ability to restore Golgi structure and localization, indicating functional differences between these mutants. To better reflect the regulatory role of MARK2-mediated phosphorylation of CAMSAP2, we performed scratch wound-healing experiments in MARK2 KO cells by establishing stable cell lines expressing CAMSAP2 S835 mutants. These experiments allowed us to assess Golgi reorientation during wound healing and are presented in Figure 5A-C.

We also attempted to generate stable cell lines expressing GFP-CAMSAP2 and its mutants in CAMSAP2 KO cells. Unfortunately, these cells consistently failed to survive, preventing successful construction of the cell lines.

We hope these experiments and explanations address the reviewer’s concerns. We are grateful for the reviewer’s constructive feedback, which has helped us refine and improve our study.

(5) The data presented in Fig. 6A and B are not sufficient to support the authors' notion that "our observation revealed notable changes in the Golgi apparatus and microtubule network distribution in relation to the wounding. (page 11)"Fig. 6A, which includes only a single-cell image in each panel, does not demonstrate the general state of microtubules and the Golgi in the wound-edge cells. The reader cannot even know the migration direction of each cell.Fig.6 B are not suitable to quantitatively support the authors' claim. The authors should find a way to quantitatively estimate the microtubule density around the Golgi and the shape and compactness of the Golgi in each cell facing the wound, not estimating the colocalization of microtubules and the Golgi, as in the present Fig. 6B.

We sincerely apologize for the confusion caused by our unclear descriptions and presentation.

Here, we clarify the purpose and improvements made to address the reviewer’s concerns. In this study, we primarily aimed to observe the relationship between microtubules and the Golgi apparatus in cells at the leading edge of the wound during directed migration. In Figure 6A (now Supplementary Figure 6E), the images represent cells located at the wound edge at different time points. To improve clarity, we have added arrows indicating the migration direction and updated the figure legend to describe these details (page 40 lines 13-14).

To better quantify the relationship between microtubules and the Golgi apparatus, we revised our analysis by referring to the quantitative method used in Figure 3F of the paper Molecular Pathway of Microtubule Organization at the Golgi Apparatus. Specifically, we performed a radial analysis of fluorescence intensity in cells at the wound edge, measuring the distance from the Golgi center (x-axis) and the normalized radial fluorescence intensity of microtubules and the Golgi (y-axis). These results are now presented in Supplementary Figure 6E and 6F.

We hope these improvements address the reviewer’s concerns and provide stronger evidence for the changes in the Golgi apparatus and microtubule network distribution in relation to wound healing. We greatly appreciate the reviewer’s constructive feedback, which has significantly enhanced the clarity and rigor of our study.

The legends to Fig. 6A and B indicate that they compared immunofluorescent staining of cells at the edge of the wound after 0.5h and 2 h of migration. However, the authors state in the text that they compared "the cells located before the wound" and "the cells at the trailing edge of the wounding (page 11)."Although this description is highly ambiguous and misleading, if they compared the wound-edge cells and the cells separated from the wound edge at 2 h after cell migration here, they should improve the experimental design as I pointed out in the 2nd major comment.

We thank the reviewer for their detailed feedback regarding the experimental design and the need to clarify our descriptions. We have addressed these concerns as follows:

- Clarification of descriptions:

We recognize that the previous description in the text regarding "the cells located before the wound" and "the cells at the trailing edge of the wounding" was ambiguous and potentially misleading. We have revised this text to accurately describe the experimental design. Specifically, we compared cells at the leading edge of the wound at different time points (0.5h and 2h post-migration). These corrections are reflected in figure legends (Supplementary Figure 6E and 6F) and the Results section (page 11,lines 3-8).

- Improved experimental design:

To better support our conclusions, we performed live-cell imaging to observe the dynamic changes in the Golgi apparatus during directed migration. As shown in Supplementary Figure 2A, our results confirm that the Golgi apparatus undergoes a transient dispersed state before reorganizing into an intact structure.

Additionally, we performed fixed-cell staining at different time points to analyze the colocalization of CAMSAP2 with the Golgi apparatus in cells at the leading edge of the wound. The colocalization analysis, presented in Figures 1A-C, further demonstrates the dynamic regulation of CAMSAP2 during Golgi reorientation.

We hope these updates address the reviewer’s concerns and provide a clearer and more robust foundation for our conclusions. We are grateful for the reviewer’s constructive feedback, which has greatly enhanced the clarity and rigor of our study.

Minor comments(1) In Fig. 2 and Supplementary Fig. 3, the authors claim that MARK2 is enriched around the Golgi. However, this claim was based on immunofluorescent images of single cells and single-line scans.It is better to present the statistical data for Pearson's coefficient as shown in Figs. 1D and E. To demonstrateMARK2 enrichment around Golgi, but not localization in Golgi, the authors should find a way to quantify the specific enrichment of MARK2 signals in the Golgi region.

We thank the reviewer for raising this important point regarding the enrichment of MARK2 around the Golgi apparatus. Upon further consideration, we acknowledge that our current data do not provide sufficient evidence to fully elucidate the mechanism of MARK2 localization to the Golgi.

To maintain the scientific rigor of our study, we have removed this claim and the corresponding content from the manuscript, including original Figures 2 and Supplementary Figure 3 that specifically discuss MARK2 enrichment. These changes do not affect the primary conclusions of the study, which focus on the role of MARK2-mediated phosphorylation of CAMSAP2.

We hope this clarification addresses the reviewer’s concerns. In the future, we plan to investigate the precise mechanism of MARK2 localization using additional experimental approaches. We are grateful for the reviewer’s constructive feedback, which has helped us refine the scope and focus of our manuscript.

(2) In Fig. 3 and Supplementary Fig. 4, the authors report that CAMSAP2 localization on the Golgi is reduced in cells lacking MARK2.Essentially, the present results support this claim. However, the authors should analyze the Golgi localization of CAMASP2 with the same quantification parameter because they used Pearson's coefficient in Fig. 1D, E and Supplementary Fig.4D but Mander's coefficient in Fig. 3C and Fig.4F.

We thank the reviewer for their insightful comment regarding the consistency of quantification parameters used in our analysis of CAMSAP2 localization on the Golgi apparatus.

To address this concern, we have revised Figure 3C to use Pearson’s coefficient for consistency with Figure 1D, 1E (Figure 1B and 1E in the revised manuscript), and Supplementary Figure 4D (Supplementary Figure 3I in the revised manuscript). This ensures uniformity in the quantification parameters across these analyses.

For Figure 4F, we have retained Mander’s coefficient, as it accounts for variability in expression levels due to overexpression in individual cells. We believe this approach provides a more accurate reflection of CAMSAP2 localization under the experimental conditions shown in Figure 4F.

We hope these adjustments clarify our analysis and address the reviewer’s concerns. We greatly appreciate the reviewer’s constructive feedback, which has helped improve the consistency and accuracy of our study.

(3) In Fig.4D-F, the authors claim that S835 phosphorylation of CAMSAP2 is essential for its localization to the Golgi apparatus and for restoring the Golgi dispersion induced by CAMASAP2 depletion.Fig.4E indicates that the S835A mutant of CAMSAP2 significantly restores the compact assembly of the Golgi apparatus, and the differences in the rescue activities of the wild type, S835A, and S835D are rather small. These data contradict the authors' conclusions regarding the pivotal role of MARK2-mediated phosphorylation at the S835 site of CAMSAP2 in maintaining the Golgi architecture (page 9). The authors should remove the phrase "MARK2-mediated" from the sentence unless addressing the aforementioned issues (see 3rd major comment) and describe the role of S835 phosphorylation in more subdued tone.

We thank the reviewer for their constructive feedback regarding the conclusions drawn about the role of MARK2-mediated phosphorylation of CAMSAP2 at S835.

In response, we have revised the relevant sentence to reflect a more nuanced interpretation of the data. Specifically, the original statement:

“These observations indicate that the phosphorylation of serine 835 in CAMSAP2 is essential for its proper localization to the Golgi apparatus.”

has been updated to:

“These observations indicate that MARK2 phosphorylation of serine at position 835 of CAMSAP2 affects the localization of CAMSAP2 on the Golgi and regulates Golgi structure” (page 9, Lines 27-29).

We hope this modification addresses the reviewer’s concerns. We are grateful for the feedback, which has helped us refine our conclusions and enhance the clarity of our manuscript.

(4) In Figs. 5I, J and Supplementary Fig.7A-E, the authors claim that the S835 phosphorylationdependent interaction of CAMSAP2 with Uso1 is essential for its localization to the Golgi apparatus.This claim was made based on immunofluorescent images of single cells and single-line scans, and was not sufficiently verified (Supplementary Fig.7B, C). Because this is a crucial claim for the present paper, the authors should present statistical data for Pearson's coefficient, as shown in Fig. 1D and E, to quantitatively estimate the Golgi localization of CAMSAP2.

We thank the reviewer for their suggestion to present statistical data using Pearson's coefficient for a more robust quantification of the Golgi localization of CAMSAP2.

In response, we have revised the statistical analysis for Supplementary Figures 7B-C (Revised Figures 6F and 6G) to use Pearson's coefficient. This change ensures consistency with the quantification methods used in Figures 1D and 1E (Revised Figures 1B and 1E), allowing for a more standardized evaluation of CAMSAP2’s localization to the Golgi apparatus.

We hope this modification addresses the reviewer’s concerns and strengthens the quantitative support for our claims. We are grateful for the reviewer’s constructive feedback, which has helped improve the rigor of our study.

(5) The signal intensities of the immunofluorescent data in Fig. 4D, Fig. 5A, Sup-Fig. 3C and E, and Sup-Fig. 7S are very weak for readers to clearly estimate the authors' claims. They should be improved appropriately.

We thank the reviewer for highlighting the need to improve the clarity of the immunofluorescent data presented in several figures.

In response, we have enhanced the signal intensities in Figures 4D, 5A, and Supplementary Figure 7D (Revised Supplementary Figure 6A) to make the signals clearer for readers, while ensuring that the adjustments do not alter the integrity of the original data. Supplementary Figures 3C and 3E was remove from our manuscript.

Additionally, to improve consistency and readability across the manuscript, we have standardized the quantification methods for similar analyses:

For CAMSAP2 localization to the Golgi, Pearson's coefficient has been used throughout the manuscript. Figure 3C has been updated to use Pearson's coefficient for consistency.

For Golgi state analysis in wound-edge cells, we have used the Golgi position relative to the nucleus as a uniform metric. This has been applied to Supplementary Figures 1F and 1G, Figures 2D and 2E, and Figures 5A and 5B.

We hope these adjustments address the reviewer’s concerns and improve the clarity and consistency of our study. We greatly appreciate the reviewer’s constructive feedback, which has significantly enhanced the quality of our manuscript.

(6) As indicated above, the authors frequently change the parameters or methods for quantifying the same phenomena (for example, the localization of CAMSAP on the Golgi and Golgi state in wound edge cells) in each figure. This is highly confusing. They should unify them.

We thank the reviewer for their valuable feedback regarding the inconsistency in quantification methods across the manuscript.

To address this concern, we have carefully reviewed the entire manuscript and standardized the methods used for quantifying similar phenomena:

- CAMSAP2 localization on the Golgi:

Pearson's coefficient is now consistently used throughout the manuscript. For example, Figure 3C has been updated to use Pearson's coefficient to align with other figures, such as Figures 1B and 1E.

- Golgi state in wound-edge cells:

The Golgi state is now uniformly measured based on the position of the Golgi relative to the nucleus. This method has been applied to Supplementary Figures 1F and 1G, Figures 2D and 2E, and Figures 5A and 5B.

We believe these changes significantly improve the clarity and consistency of the manuscript, ensuring that readers can easily interpret the data. We are grateful for the reviewer’s constructive feedback, which has greatly helped us enhance the quality and rigor of our study.

(7) The legends frequently fail to clearly indicate the number of independent experiments on which each statistical analysis was based.

We thank the reviewer for highlighting the need to clearly indicate the number of independent experiments for each statistical analysis.

In response, we have carefully reviewed the entire manuscript and updated the figure legends to include the number of independent experiments for every statistical analysis. This ensures transparency and allows readers to better evaluate the reliability of the data.

We hope these updates address the reviewer’s concerns and improve the clarity and rigor of the manuscript. We appreciate the reviewer’s constructive feedback, which has helped us enhance the quality of our work.

(8) Supplemental Figs. 4E and 4F are not cited in the text.

We thank the reviewer for pointing out that Supplemental Figures 4E and 4F were not cited in the text.

To address this, we have updated the manuscript to cite these figures (Revised Figures 2H and 2I) in the appropriate section (page 8, lines 1-5).

“the absence of MARK2 can also influence the orientation of the Golgi apparatus during cell wound healing and cause a delay in wound closure (Figure 2 D-I and Figure 3 D).”

We hope this revision resolves the reviewer’s concern and improves the clarity and completeness of the manuscript. We appreciate the reviewer’s feedback, which has helped us refine our work.

(9) The data in Fig. 3 analyzed MARK2 knockout cells (not knockdown cells). The caption should be corrected.

We thank the reviewer for pointing out the incorrect use of "knockdown" in the caption of Figure 3.

To address this, we have revised the title of Figure 3 from:

“MARK2 knockdown reduces CAMSAP2 localization on the Golgi apparatus.”

to:

“MARK2 affects CAMSAP2 localization on the Golgi apparatus.”

This updated caption reflects the inclusion of both MARK2 knockout and knockdown cell lines analyzed in Figure 3.

We hope this correction resolves the reviewer’s concern and ensures the accuracy of our manuscript. We greatly appreciate the reviewer’s attention to detail, which has helped us improve the clarity and consistency of our work.

(10) The present caption in Fig. 6 disagrees with the content of the figure.

We thank the reviewer for pointing out the inconsistency between the caption and the content of Figure 6.

To address this issue, we have revised the content of Figure 6 to ensure it aligns accurately with the caption. The updated figure now reflects the description provided in the caption, eliminating any discrepancies and improving clarity for the readers.

We appreciate the reviewer’s constructive feedback, which has helped us enhance the accuracy and presentation of our manuscript.

(11) What do "CS" indicate in Fig. 4B and Supplementary Fig. 5D? The style used to indicate point mutants of CAMSAP2 should be unified. 835A or S835A?

We thank the reviewer for pointing out the inconsistency in the naming of CAMSAP2 mutants.

To address this, we have revised all relevant figures and text to use the consistent format "S835A" and "S589A" for CAMSAP2 mutants. Specifically, in Figure 4B and Supplementary Figure 5D (now Supplementary Figure 4C), we have replaced the abbreviation "CS2" with "CAMSAP2" and updated the mutant names from "835A" and "589A" to "S835A" and "S589A," respectively. We hope these updates resolve the reviewer’s concerns and ensure clarity and consistency throughout the manuscript. We are grateful for the reviewer’s attention to detail, which has helped us improve the quality of our work.

(12) Uso1 is not a Golgi matrix protein.

We thank the reviewer for pointing out the incorrect description of Uso1 as a Golgi matrix protein.

In response, we have revised the manuscript to replace all references to “USO1 as a Golgi matrix protein” with “USO1 as a Golgi-associated protein.” This correction ensures that the terminology used in the manuscript is accurate and consistent with current scientific understanding.

We appreciate the reviewer’s attention to detail, which has helped us improve the accuracy and quality of our manuscript.